# When does predictive inverse dynamics outperform behavior cloning? Exploring the role of action and state uncertainty

## Abstract

Offline imitation learning aims to train agents from demonstrations without interacting with the environment, but standard approaches like behavior cloning (BC) often fail when expert demonstrations are limited. Recent work has introduced a class of architectures we call predictive inverse dynamics models (PIDM), which combine a future state predictor with an inverse dynamics model (IDM) to infer actions to reach the predicted future states. While PIDM often outperforms BC, the reasons behind its benefits remain unclear. In this paper, we analyze PIDM in the offline imitation learning setting and provide a theoretical explanation: conditioning the IDM on the predicted future state reduces variance, whereas predicting the future state introduces bias. We establish conditions on the state predictor bias for PIDM to achieve lower prediction error and higher sample efficiency than BC, with the gap widening when additional data sources are available. The efficiency gain is characterized by the variance of actions conditioned on future states, highlighting PIDM's ability to reduce uncertainty in states where future context is informative. We validate these insights empirically under more general conditions in 2D navigation tasks using human demonstrations, where BC requires up to five times (three times on average) more samples than PIDM to reach comparable performance. Finally, we extend our evaluation to a complex 3D environment in a modern video game with high-dimensional visual inputs and stochastic transitions, showing BC requires over 66% more samples than PIDM in a realistic setting.

## 1 Introduction

Offline imitation learning aims to learn closed-loop control policies that replicate expert behavior using only pre-collected data, without access to a reward function or further interaction with the environment. This paradigm has broad applicability across domains such as robotics (Schaal, 1999; Fang et al., 2019), autonomous driving (Pan et al., 2020), and gaming (Pearce & Zhu, 2022; Pearce et al., 2023; Schäfer et al., 2023). A prominent line of research in imitation learning focuses on one- or few-shot generalization, where models are pretrained on large-scale datasets spanning diverse tasks (Duan et al., 2017), with the goal of adapting to new tasks from only a handful of demonstrations. However, collecting such large-scale expert demonstrations is often costly, time-consuming, or infeasible—particularly in specialized domains like robotics, where data acquisition is expensive and task-specific. As a result, many real-world applications lack the scale of data required to train or adapt large models using standard imitation learning techniques.

In contrast to approaches that rely on extensive pretraining, we focus on the low-data regime, where only few demonstrations are available for the target task, and no additional data can be assumed. This setting is increasingly relevant in the current AI landscape, where large foundation models are trained on massive datasets, yet aligning them to new domains with limited supervision remains a significant challenge.

The most common offline imitation learning approach is behavior cloning (BC) (see Figure 2a), which can exhibit complex behavior (Osa et al., 2018; Pearce & Zhu, 2022; Florence et al., 2022) but typically relies on the availability of many demonstrations per task. Recent work has introduced a promising alternative to BC, which we refer to as predictive inverse dynamics models (PIDMs) (Du

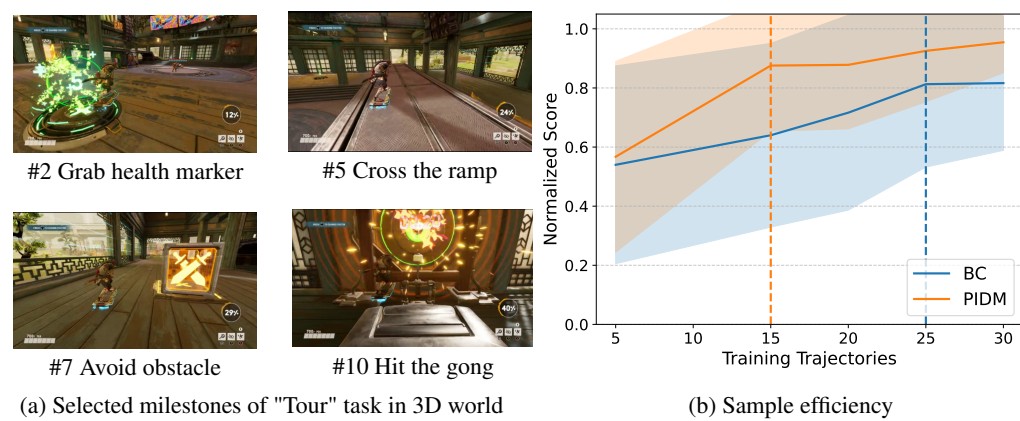

| #2 Grab health marker | #5 Cross the ramp |
| #7 Avoid obstacle | #10 Hit the gong |

(a) Selected milestones of "Tour" task in 3D world

(b) Sample efficiency

Figure 1: (a) Visualization of selected milestones from the complex "Tour" task in a 3D world with video input, stochastic transitions, and real-time inference. (b) Sample efficiency curves (mean $\pm$ std) for PIDM and BC. BC requires 66% more samples to achieve 80% success rate on average.

et al., 2023; Xie et al., 2025; Tian et al., 2025). PIDM integrates two components: a state predictor, which forecasts plausible future states, and an inverse dynamics model (IDM), which infers the actions needed to reach those states (see Figure 2d). This modular design offers a key advantage—it allows leveraging diverse data sources, including action-free demonstrations and non-expert data. By augmenting a small set of expert demonstrations with such additional data, PIDM has demonstrated strong empirical performance (Xie et al., 2025). Interestingly, Xie et al. (2025) also reported that PIDM can significantly improve upon BC even when no additional data sources were available, suggesting the promise of PIDM for the low-data regime. However, the underlying reasons for their sample efficiency remain unclear. Is there something intrinsic to the PIDM architecture that enables this advantage? Under what conditions can we expect such gains to consistently emerge?

In this work, we analyze PIDM and provide theoretical insights into why decomposing the decision-making problem into a state predictor and an IDM can lead to significant sample efficiency improvements over BC. Specifically, PIDM can achieve comparable or superior performance using fewer expert demonstrations. First, we show that the prediction error of an optimal estimator for PIDM is always less than or equal to that of BC, resulting in a non-negative performance gap in favor of PIDM even in the small-data regime. This gap is characterized by the expected conditional variance of actions given all possible future states. We then extend the analysis to arbitrary estimators and highlight a key advantage of the PIDM architecture: a bias-variance tradeoff. Conditioning on a future state reduces the total variance by removing the conditional variance mentioned above; however, predicting a future state induces bias, reducing the effective gap. This bias-variance tradeoff also appears when comparing the sample efficiency of BC and PIDM, and we provide conditions on the state predictor bias for PIDM to be at least as sample-efficient than BC.

Second, we provide empirical evidence that the predicted sample efficiency gains apply to more general conditions, including the small-data regime, with no additional data sources, and general modeling approaches, like neural networks. We perform experiments on a benchmark of four 2D navigation tasks in a state-based environment, using a dataset of human demonstrations, and observe that BC requires between $1.3\times$ and $4\times$ more demonstrations than PIDM. The simplicity of the environment allows us to understand how the theory works in practice by looking at the prediction error gap per state. It also allows us to isolate the efficiency gains due to the predicted error gap from the representational benefits of IDM shown in previous work (Lamb et al., 2023; Koul et al., 2023; Levine et al., 2024; Islam et al., 2022).

Finally, once we have built intuition as to *why* the PIDM decomposition is effective, we extend our investigation to complex tasks that require imitating complex navigation tasks, from image inputs, in a 3D world with stochastic transitions, in real-time using human demonstrations. In this real-world setting, sample efficiency is critical since obtaining human demonstrations is costly, and real-time requirements introduce additional constraints on the solution. In this setting, we continue to observe

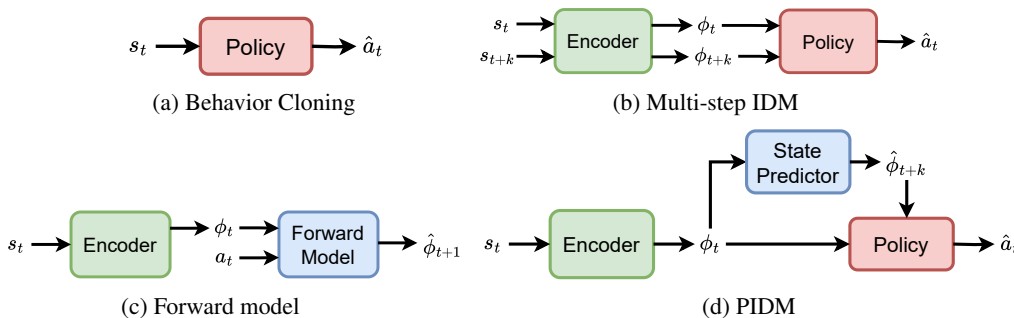

Figure 2: (a) BC learns a policy conditioned on the current state. (b) IDM learns a policy conditioned on the current and future state $k$ steps ahead. (c) Forward models predict a future state (representation) given a state and action. Both (b) IDM and (c) forward models can serve as auxiliary objectives to learn effective state representations. (d) PIDM represents an alternative to BC consisting of a state predictor, akin to an action-free forward model, that predicts future state representations, and an IDM policy. The state encoder alleviates the dependence on ground-truth future states at evaluation time.

substantial efficiency gains — BC requires 66% more samples than PIDM, demonstrating that the predicted performance gap is relevant for real-world applications.

## 2 RELATED WORK

**Inverse dynamics models.** Inverse dynamics models (IDMs) predict the initial action that initiates a sequence leading to a transition from the current state to a future state $k$ steps ahead. As a direct *imitation learning* mechanism, multi-step inverse objectives are commonly used to train policies or transferable encoders on high-dimensional observations; the inverse loss filters out exogenous factors (Mhammedi et al., 2023; Efroni et al., 2022; Lamb et al., 2023) and learns rich state representations which can later support policy learning or be reused across tasks. To enable efficient training, we focus on architectures, in which states are encoded into a latent space with a state encoder (see Figure 2b). Both the state predictor and the IDM policy operate in this latent space.

**Forward models.** Forward models (see Figure 2c) can be added as auxiliary objectives to improve learned representations (Levine et al., 2024). Alternatively, a forward model can be learned for planning purposes, e.g. using reinforcement learning (Thrun et al., 1990; Hafner et al., 2025) or model-predictive control (Zhou et al., 2024; Bar et al., 2025). They are different from the state predictor of PIDM in two ways. First, forward models require action input, while the state predictor is conditioned on the current state only. Second, forward models usually generate the next state, while the state predictor generates future states $k \geq 1$ steps ahead.

**Predictive inverse dynamics models.** Recent works have combined a state predictor with an IDM to learn generalizable policies. Inspired by diffusion models for video generation, Du et al. (2023) trained a diffusion model to predict future images conditioned on task descriptions, operating directly in image space. To simplify learning, Xie et al. (2025) proposed using compact image representations, enabling state predictors to train action-free demonstrations and IDMs on diverse, action-labeled trajectories. Tian et al. (2025) proposed end-to-end training, using the IDM objective to guide the state predictor. These approaches showed PIDM outperforms BC and other baselines. Our work provides theoretical and empirical insight into why decomposing future state and action prediction leads to this performance gain.

**Behavior cloning and trajectory modeling.** Recent analysis (Foster et al., 2024) argues that many practical implementations of BC, which rely on a log-loss, are implicitly modeling the whole state-action sequence. Our work complements such analysis by providing evidence that explicitly modeling part of the trajectory, the future state in the PIDM case, can improve sample efficiency.

## 3 PRELIMINARIES

**Problem setting.** We consider the problem setting of an MDP defined by $(\mathbb{S}, \mathbb{A}, \mathcal{T}, \mathcal{R}, d_0)$ of state space $\mathbb{S}$; action space $\mathbb{A}$; transition function $\mathcal{T} : \mathbb{S} \times \mathbb{A} \mapsto \mathcal{P}(\mathbb{S})$, where $\mathcal{P}(\cdot)$ is a probability measure; reward function $\mathcal{R} : \mathbb{S} \times \mathbb{A} \mapsto \mathbb{R}$; and initial state distribution $d_0$. To interact with the MDP, we first sample an initial state $s_0 \sim P_0$. Then, at each time step, we sample an action $a_t \sim \pi(\cdot \mid s_t)$ from the policy $\pi : \mathbb{S} \mapsto \mathcal{P}(\mathbb{A})$. Given this action, the environment transitions to a new state $s_{t+1} \sim \mathcal{T}(\cdot \mid s_t, a_t)$ and provides a reward $r = \mathcal{R}(s_t, a_t)$. Let $\mathcal{D}$ denote the resulting data distribution. For this work, we assume no access to the reward signal and consider the offline imitation learning setting, in which we are given a dataset of trajectories of states and actions generated by following some unknown expert policy $\pi^*$. Our goal is to learn a policy that is as close as possible to $\pi^*$. We consider two architectures: BC and PIDM.

**BC** treats offline imitation learning as a supervised learning problem and trains a policy to imitate the actions in the dataset given the most recent state. It consists of a single block, the policy (see Figure 2a), which can be trained to minimize the following loss between the action distribution induced by the learned policy $\pi_\mu$ and the ground truth actions under the data distribution:

$$\mathcal{L}_{\text{BC}}(\pi_\mu) = \mathbb{E}_{(\boldsymbol{s}_t, \boldsymbol{a}_t) \sim \mathcal{D}, \hat{\boldsymbol{a}}_t \sim \pi_\mu(\cdot \mid \boldsymbol{s}_t)} \left[ \ell(\hat{\boldsymbol{a}}_t, \boldsymbol{a}_t) \right] \tag{1}$$

for some dissimilarity measure, denoted generically as $\ell$. There are multiple choices on how BC approximates the policy distribution that offer different fidelity and complexity tradeoffs, ranging from simple but effective point estimates to rich but complex generative models that can capture distributions with multiple modes.

**PIDM** consists of two main submodels (see Figure 2d): a state predictor, $p$, that predicts future states for some horizon $k$, and an inverse dynamics model (IDM) policy, $\pi_\xi$, that predicts the next action needed to get from the current observation to the future observation in $k$ steps. They can be trained using the following losses:

$$\mathcal{L}_{\text{SP}}(p) = \mathbb{E}_{(\boldsymbol{s}_t, \boldsymbol{s}_{t+k}) \sim \mathcal{D}, \hat{\boldsymbol{s}}_{t+k} \sim p(\cdot \mid \boldsymbol{s}_t)} \left[ \ell(\hat{\boldsymbol{s}}_{t+k}, \boldsymbol{s}_{t+k}) \right], \tag{2}$$

$$\mathcal{L}_{\text{IDM}}(\pi_\xi) = \mathbb{E}_{(\boldsymbol{s}_t, \boldsymbol{a}_t) \sim \mathcal{D}, \hat{\boldsymbol{s}}_{t+k} \sim p(\cdot \mid \boldsymbol{s}_t), \hat{\boldsymbol{a}}_t \sim \pi_\xi(\cdot \mid \boldsymbol{s}_t, \hat{\boldsymbol{s}}_{t+k})} \left[ \ell(\hat{\boldsymbol{a}}_t, \boldsymbol{a}_t) \right]. \tag{3}$$

However, this is just one alternative, since PIDM offers many design choices beyond how to approximate the distributions. For instance, the state predictor and IDM can be trained jointly—as suggested by (2)–(3), with the action conditioned on the predictor's output—or they can be learned independently. Alternatively, the models might be obtained in a lazy manner, without an explicit loss function. Both the state predictor and IDM can use the same datasets or leverage different data sources. Furthermore, predictions for state and action can be conditioned directly on the input space, or an encoder can be introduced so that both submodels share a common latent space (see Figure 2d). In Section 5, we focus on the case where both submodels use the same dataset, share a common latent space, and the state-predictor is an instance-based (lazy) model.

Section 4 provides insights on the potential benefits that the PIDM architecture can provide over BC.

## 4 THEORETICAL ANALYSIS

The PIDM approach can be seen as a decomposition of BC with explicit modeling of future states:

$$\pi_\mu(a_t \mid s_t) = \int_{\mathbb{S}} p^\star(s_{t+k} \mid s_t) \pi_\xi(a_t \mid s_t, s_{t+k}) ds_{t+k}, \tag{4}$$

where $p^\star$ denotes the true future state distribution. Intuitively, this decomposition can simplify the learning of a policy whenever the conditioning on the future state in the IDM policy provides useful information to identify which action to take. In this section, we study the potential gains of PIDM over BC. All proofs are in Appendix A.

For simplicity, we consider the case where the BC and IDM policies are single-point estimators that approximate the expected action. Let $\overline{\mu}(\boldsymbol{s}_t) \triangleq \mathbb{E}[\boldsymbol{a}_t \mid \boldsymbol{s}_t]$ and $\overline{\xi}(\boldsymbol{s}_t, \boldsymbol{s}_{t+k}) \triangleq \mathbb{E}[\boldsymbol{a}_t \mid \boldsymbol{s}_t, \boldsymbol{s}_{t+k}]$ be the optimal estimators for $\pi_\mu$ and $\pi_\xi$, respectively.

Introduce the predicted error gap between the estimators of the BC and IDM policies:

$$\Delta \triangleq \text{EPE}(\overline{\mu}) - \text{EPE}(\overline{\xi}), \tag{5}$$

where $\mathrm{EPE}(\cdot)$ is the expected prediction error, which for a random variable $\boldsymbol{y}|\boldsymbol{x}$ and an estimator $\zeta(\boldsymbol{x})$ is given by: $\mathrm{EPE}(\zeta) \triangleq \mathbb{E}_{\boldsymbol{x}}\left[(\boldsymbol{y} - \zeta(\boldsymbol{x}))^2\right]$.

Our first result quantifies $\Delta$ in terms of the uncertainty in $\boldsymbol{a}_t$ due to uncertainty in $\boldsymbol{s}_{t+k}$.

**Theorem 1.** *For optimal estimators $\overline{\mu}$ and $\overline{\xi}$, The predicted error gap is given by:*

$$\Delta = \mathbb{E}_{\boldsymbol{s}_t}\left[\mathrm{Var}_{\boldsymbol{s}_{t+k}|\boldsymbol{s}_t}\left(\mathbb{E}\left[\boldsymbol{a}_t \mid \boldsymbol{s}_t, \boldsymbol{s}_{t+k}\right]\right)\right] \geq 0. \tag{6}$$

Theorem 1 shows that knowing $\boldsymbol{s}_{t+k}$ can increase the prediction accuracy of $\boldsymbol{a}_t$. However, this improvement assumes access to the exact state predictor distribution. When we only have access to an approximate state predictor, denoted $\widehat{p}$, the estimator of the IDM policy will generate actions conditioned on samples of the form $(s_t, s'_{t+k})$ with $s'_{t+k} \sim \widehat{p}(\cdot \mid s_t)$. This distribution shift ($p^\star \neq \widehat{p}$) introduces bias. In other words, the estimator of the PIDM policy has two sources of bias: the bias that is intrinsic to the IDM policy estimator and the additional bias due to the distribution shift of the state predictor. This additional bias reduces the predicted error gap, as shown in the following corollary that extends Theorem 1 to any (non-optimal) estimator. Let $\mathbb{E}_{\mathcal{D}_n}[\cdot]$ denote expectation over datasets with i.i.d samples from the true data distribution, i.e., $\mathcal{D}_n \triangleq \left\{(s_t, a_t, s_{t+k}) \overset{\text{i.i.d.}}{\sim} \mathcal{D}\right\}_{t=1}^n$; while $\mathbb{E}_{\mathcal{D}_{\widehat{p},m}}[\cdot]$ emphasizes expectation over datasets under an approximate state predictor, i.e., $\mathcal{D}_{\widehat{p},m} \triangleq \left\{(s_t, a_t, s'_{t+k}) \mid (s_t, a_t) \overset{\text{i.i.d.}}{\sim} \mathcal{D}, s'_{t+k} \sim \widehat{p}(\cdot \mid s_t)\right\}_{t=1}^m$. The i.i.d. assumption simplifies the analysis and is a valid approximation for fast mixing chains. Introduce also shortcuts for the bias:

$$b_\mu^2(\widehat{\mu}) \triangleq \mathbb{E}_{\boldsymbol{s}_t}\left[\left(\mathbb{E}_{\mathcal{D}_n}\left[\widehat{\mu}(\boldsymbol{s}_t)\right] - \overline{\mu}(\boldsymbol{s}_t)\right)^2\right], \tag{7}$$

$$b_\xi^2(\widehat{\xi_p}) \triangleq \mathbb{E}_{\boldsymbol{s}_t, \boldsymbol{s}_{t+k}}\left[\left(\mathbb{E}_{\mathcal{D}_{\widehat{p},m}}\left[\widehat{\xi_p}(\boldsymbol{s}_t, \boldsymbol{s}_{t+k})\right] - \overline{\xi}(\boldsymbol{s}_t, \boldsymbol{s}_{t+k})\right)^2\right]. \tag{8}$$

**Corollary 1.** *Let $\widehat{\mu}$ and $\widehat{\xi_{\widehat{p}}}$ be the estimator of the BC and IDM policies obtained with $\mathcal{D}_n$ and $\mathcal{D}_{\widehat{p},m}$, respectively. Let the difference in the estimators' own variance and bias be given by:*

$$\delta \triangleq \mathbb{E}_{\boldsymbol{s}_t}\left[\mathrm{Var}\left(\widehat{\mu}\left(\boldsymbol{s}_t\right)\right)\right]] - \mathbb{E}_{\boldsymbol{s}_t, \boldsymbol{s}_{t+k}}\left[\mathrm{Var}\left(\widehat{\xi_{\widehat{p}}}\left(\boldsymbol{s}_t, \boldsymbol{s}_{t+k}\right)\right)\right], \tag{9}$$

$$\beta \triangleq b_\mu^2(\widehat{\mu}) - b_\xi^2(\widehat{\xi_{\widehat{p}}}). \tag{10}$$

*And let $\Delta$ be given by (6). Then, the predictor error gap is given by:*

$$\widehat{\Delta}_{\widehat{p}} \triangleq \mathrm{EPE}\left(\widehat{\mu}\right) - \mathrm{EPE}\left(\widehat{\xi_{\widehat{p}}}\right) = \Delta + \delta + \beta. \tag{11}$$

Corollary 1 shows how the PIDM architecture introduces a bias-variance tradeoff: $\Delta$ represents the variance reduction of the IDM policy due to knowing the future state; while $\beta \leq 0$ represents the additional bias induced by an approximate state predictor, assuming both estimators have similar intrinsic bias (as required for a fair comparison). Corollary 1 also motivates the use of two additional data sources. First, by using additional action-free demonstrations of the same task to train a more accurate state predictor model, we can reduce the bias due to $\widehat{p}$ and make $\beta \to 0$. Second, additional expert demonstrations from different tasks in the same environment, or even non-expert demonstrations when $k = 1$, can be used to reduce the variance of $\widehat{\xi_{\widehat{p}}}$ and make $\delta > 0$.

The next results connect the prediction error gap with sample efficiency gains. We assume asymptotic efficiency for simplicity, which implies that the MSE of the estimator decreases approximately linearly with the number of samples. Let $F_\mu$ and $F_\xi$ denote the Fisher information for $\pi_\mu$ and $\pi_\xi$, respectively. Let $\gtrsim$ denote greater than or approximately equal to.

**Theorem 2.** *Let $\widehat{\mu}_n$ and $\widehat{\xi_{\widehat{p}},m}$ be asymptotically efficient estimator of the BC and IDM policies obtained with $\mathcal{D}_n$ and $\mathcal{D}_{\widehat{p},m}$, respectively, where $n$ and $m$ denote the minimum number of samples required to achieve error level $\varepsilon$. Let $F_\mu$ and $F_\xi$ exist, and let $\pi_\xi$ satisfy regularity conditions (for differentiating under the integral sign). Then, for large enough $n$ and $m$, we have:*

$$\eta \triangleq \frac{n}{m} \approx \frac{F_\xi}{F_\mu} \frac{\left(\frac{\partial}{\partial \mu} b_\mu(\widehat{\mu}_n) + 1\right)^2}{\left(\frac{\partial}{\partial \xi} b_\xi(\widehat{\xi_{\widehat{p}},m}) + 1\right)^2} \left(1 + \frac{\Delta + b_\mu^2(\widehat{\mu}_n) - b_\xi^2(\widehat{\xi_{\widehat{p}},m})}{\varepsilon - \mathbb{E}_{\boldsymbol{s}_t}\left[\mathrm{Var}(\boldsymbol{a}_t \mid \boldsymbol{s}_t)\right] - b_\mu^2(\widehat{\mu}_n)}\right). \tag{12}$$

Theorem 2 shows the same bias-variance tradeoff as with the EPE gap: $\Delta$ is the variance reduction that increases the sample efficiency gain and $b_\xi$ is the bias term that reduces it (by scaling down and subtraction). Based on this result, the following theorem and corollary provide conditions under which PIDM is guaranteed to be at least as sample efficiency as BC.

**Theorem 3.** *Under the conditions of Theorem 2, assume the following condition holds:*

$$b_\xi^2(\widehat{\xi}_{\widehat{p},m}) + \left(\overline{\varepsilon} - b_\mu^2(\widehat{\mu}_n)\right) \frac{\left(\frac{\partial}{\partial \xi} b_\xi(\widehat{\xi}_{\widehat{p},m}) + 1\right)^2}{\left(\frac{\partial}{\partial \mu} b_\mu(\widehat{\mu}_n) + 1\right)^2} \leq \overline{\varepsilon} + \Delta, \tag{13}$$

*where $\overline{\varepsilon} \triangleq \varepsilon - \mathbb{E}_{\boldsymbol{s}_t}\left[\mathrm{Var}(\boldsymbol{a}_t \mid \boldsymbol{s}_t)\right]$. Then: $\eta \gtrsim 1$.*

Although the result in Theorem 3 may appear complex at first glance, it is actually quite intuitive. For example, when the bias derivatives are small, Equation (13) simplifies to:

$$b_\xi^2(\widehat{\xi}_{\widehat{p},m}) - b_\mu^2(\widehat{\mu}_n) \leq \Delta. \tag{14}$$

Assuming both estimators have similar intrinsic bias (as required for a fair comparison), Equation (14) implies that for PIDM to be more sample efficient than BC, the bias induced by an approximate state predictor must not exceed the variance reduction achieved by conditioning on the future state. Given this intuition, the following result follows naturally.

**Corollary 2.** *Under the conditions of Theorem 2, if $\widehat{\xi}_{\widehat{p},m}$ is asymptotically unbiased, then: $\eta \gtrsim 1$.*

Although these conditions have been derived for large enough samples, they rely on the asymptotic covariance, which is often a good predictor of finite-time performance (see discussion in Appendix A.6). Indeed, Section 5 provides empirical evidence that efficiency gains hold even in the small-data regime. Furthermore, experiments in Appendix D.3 also confirm the predicted role of the state predictor bias under limited data.

## 5 EXPERIMENTS

To better understand how the EPE efficiency gains predicted in Section 4 manifest in task completion efficiency gains in practice, we perform experiments in a 2D navigation environment, where we can easily analyze the properties of datasets and policies. We then conduct experiments in a 3D world that require precise execution of a complex task from images to validate our findings under real-world conditions.

### 5.1 ENVIRONMENTS

**2D navigation environment.** We consider four tasks of varying complexity within a 2D navigation environment, visualized in Figure 3, in which the agent needs to reach a sequence of goals. The tasks are fully observable with low-dimensional states containing the x- and y-position of the agent as well as the positions of all goals, and whether they have already been reached. This simplified setting allow us to study the efficiency gains of PIDM over BC due to its action decomposition, isolated from other gains resulting from improved representations reported in prior work (Lamb et al., 2023; Koul et al., 2023; Levine et al., 2024). The agent chooses actions in $[-1, 1]^2$ for its movement, and the transitions are stochastic with Gaussian noise $\mathcal{N}(0, 0.2)$ added to the actions. For each task, a human player collected a dataset of 50 trajectories by navigating the agent to reach all goals using a controller. The datasets naturally contain some variability in actions in any given state as visualized by the human trajectories shown in Figure 3. We refer to Appendix B for more details on each human dataset. To identify the impact of the action variability in the data collection policy on BC and PIDM, we conduct further experiments in the same tasks using datasets collected with a deterministic A$^*$ planner policy (see Appendix D).

**3D world.** For a complex environment under real-world conditions, we constructed a dataset comprising human gameplay demonstrations within a modern 3D video game titled "Bleeding Edge", developed by Ninja Theory. The environment corresponds to the "Dojo" practice level. It features a third-person perspective with a freely controllable camera, where the camera orientation

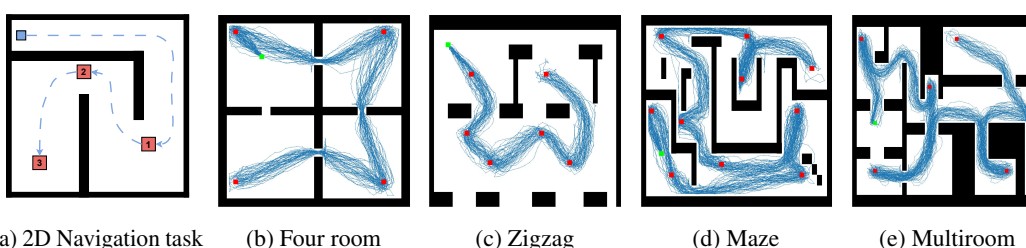

(a) 2D Navigation task    (b) Four room    (c) Zigzag    (d) Maze    (e) Multiroom

Figure 3: Visualization of 2D navigation environment. (a) Tasks require the agent (blue box) to navigate to reach the goals (red boxes) in a particular order. (b) - (e) Visualizations of all four tasks and the traces of the 50 trajectories within the datasets.

directly affects the agent's movement direction, introducing a non-trivial perception-action coupling. Observations are captured as raw video frames, which are processed through a pre-trained image encoder to obtain embeddings that facilitate efficient learning. These embeddings are subsequently passed to the networks of the algorithms. In our experiments, we use the pre-trained ViT-B/16 Theia vision encoder (Shang et al., 2024). The action space contains continuous actions $[-1, 1]^4$ to control the x- and y-movement of the controlled character and the camera. State transitions occur asynchronously at 30 FPS and require real-time inference. Due to the game's deployment on a remote server in a distant cloud region, transitions are stochastic, affected by variable latency and visual artifacts. Within the environment we consider a task we refer to as "Tour" that consists of ~36 seconds of precise navigation with 11 milestones, testing the agents' capability to steer and stay on track while avoiding obstacles and reacting at objects of interests (see Figure 1a for visualization of some milestones and Appendix C.2 for the complete list).

### 5.2 Algorithms

**Model architecture.** In the 2D navigation environment with fully observable states, we train MLP networks for the encoder and policy networks of BC and PIDM, and use $k = 1$ for PIDM. In contrast, the complex 3D world task is partially observable with inputs being video frames that are first being processed by a pre-trained vision encoder. The policy then receives a stack of vision encoder embeddings for three frames spanning one second to approximate a single state. BC and PIDM policies are then conditioned on these stacked representations for the current state and, in the case of PIDM, for the future state. To leverage the representational benefits of multi-step IDM (Lamb et al., 2023; Koul et al., 2023), we train the PIDM policy using $k \in \{1, 6, 11, 16, 21, 26\}$ for this task and additionally condition the PIDM policy network on a one-hot encoding of $k$. During evaluation, we query the PIDM policy and state predictor with $k = 1$. For BC and PIDM, we use the $tanh$ activation function on the action logits to get actions in the desired $[-1, 1]$ range.

**State predictor.** We consider two state predictors. In the 2D navigation environment, we leverage an instance-based learning model (Keogh, 2010) for a deterministic state predictor:

$$p(s_t) = s_{\tau^\star + k}^{i^\star} \quad \text{with} \quad (\tau^\star, i^\star) \triangleq \arg\min_{\tau, i} ||s_t - s_\tau^i||^2, \tag{15}$$

with $s_\tau^i$ referring to the state at time step $\tau$ in demonstration $i$ of the training dataset. In short, the state predictor first queries for the nearest state within any training demonstration, as measured by the Euclidean distance, and then predicts the state $k$ steps ahead of that state within the same training demonstration. In the 2D navigation environment, we further constrain the query for the nearest state to only match states in which the same goal needs to currently be reached. Computation of this lookup is efficient for the small-data regime considered in our work.

In the complex 3D world task, we consider a simplified state predictor that is conditioned on the time step $t$ and a single training demonstration, denoted with superscript $i$, and returns the state $s_{t+k}^i$ within that training demonstration. Despite its simplicity, we find that even this simple state predictor can enable effective evaluation when paired with an IDM model.

**Training details.** To train the policies, we sample batches of 4096 $(s_t, a_t)$ or $(s_t, a_t, s_{t+k})$ tuples for BC and IDM policies, respectively, using ground-truth states and actions from training demon-

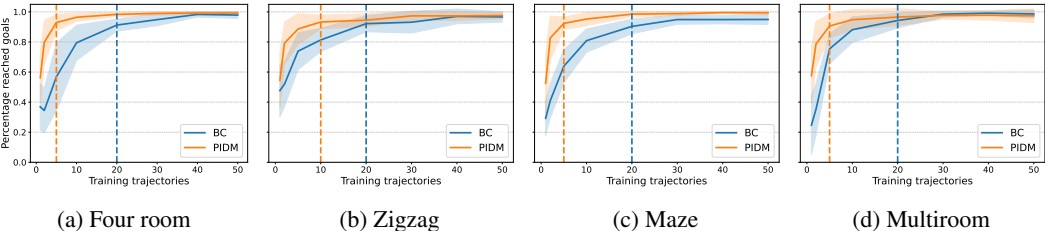

(a) Four room       (b) Zigzag       (c) Maze       (d) Multiroom

Figure 4: Performance per number of training demonstrations for BC and PIDM in four tasks trained on human datasets. Lines and shading correspond to the average and standard deviation across 20 seeds. We further visualize the number of samples required by PIDM and BC to reach 90% of the highest achievable performance with vertical dotted lines.

Table 1: Maximum reached goal ratio and sample efficiency ratios of PIDM over BC for 2D navigation tasks and average across tasks.

| Task | Four room | Zigzag | Maze | Multiroom | Average |
|---|---|---|---|---|---|
| max BC ↑ | 0.98 | 0.97 | 0.95 | 0.99 | – |
| max PIDM ↑ | 0.99 | 0.98 | 0.99 | 0.98 | – |
| $\eta_{\text{PIDM}}(80\%)$ ↑ | 4.0 | 2.0 | 5.0 | 2.0 | 3.25 |
| $\eta_{\text{PIDM}}(90\%)$ ↑ | 4.0 | 2.0 | 4.0 | 4.0 | 3.5 |
| $\eta_{\text{PIDM}}(95\%)$ ↑ | 4.0 | 1.33 | 5.0 | 1.5 | 3.00 |

strations. All networks are optimized end-to-end for $100\,000$ optimization steps from the BC and IDM losses defined in Equation (1) and Equation (3) using the Adam optimizer. For further details on hyperparameter tuning and architectures used in the 2D navigation and complex video game environments, please refer to Appendix B and Appendix C, respectively.

## 5.3 SAMPLE EFFICIENCY GAINS FOR 2D NAVIGATION

To study the sample efficiency gains of PIDM, we train a BC and PIDM on each dataset with varying numbers of trajectories, namely (1, 2, 5, 10, 20, 30, 40, 50). Our performance metric is the fraction of reached goals in the right order. For each task and number of training demonstrations, we train BC and PIDM for 20 random seeds, and evaluate four checkpoints throughout training of each seed using 50 rollouts. We report aggregate results over the average performance of 20 seeds for the best checkpoint for each task and number of training demonstrations.

To summarize efficiency gains, we compute efficiency ratios $\eta_{\text{PIDM}}$ for each task, given by

$$\eta_{\text{PIDM}}(c) = \frac{n(\text{BC}, c)}{n(\text{PIDM}, c)}, \tag{16}$$

where $n(\text{A}, x)$ is the average number of samples required by algorithm A to reach at least a fraction $c$ (expressed as a percentage) of the task's maximum attainable performance. In other words, we compute the ratio of the number of samples needed by BC and PIDM to obtain similar performance. Figure 4 visualizes the percentage of reached goals for BC and PIDM across varying number of samples, and Table 1 summarizes efficiency ratios, showing significant sample efficiencies for IDM over BC, as predicted by the analysis in Section 4. We find BC requires up to $5\times$ more demonstrations than PIDM to achieve comparable performance, and $3\times$ on average across tasks. When training on less diverse demonstrations collected by a deterministic A$^*$ planner, we find that the sample efficiency gains of PIDM over BC are further amplified, as shown in Appendix D.

## 5.4 FUTURE CONDITIONING AS A VARIANCE REDUCTION OPPORTUNITY

Our theoretical insights of Theorem 1 and Theorem 2 indicate that states, in which there is high uncertainty on the action due to uncertainty in the future state, as given by $\Delta(s)$, are key to potential

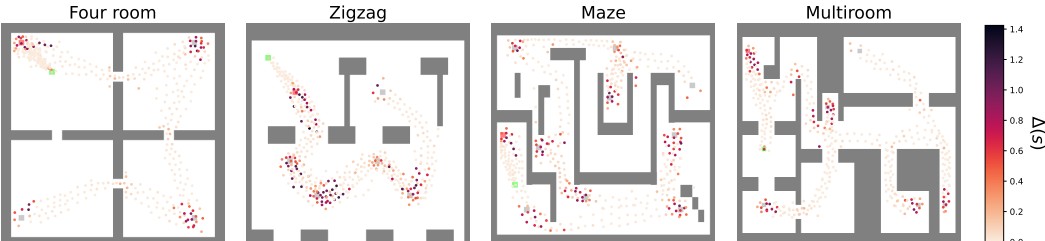

Figure 5: Visualized state-wise EPE gaps $\Delta(s)$ from Equation (17) computed for each dataset. We observe large gaps in states surrounding the goals where human actions are more diverse.

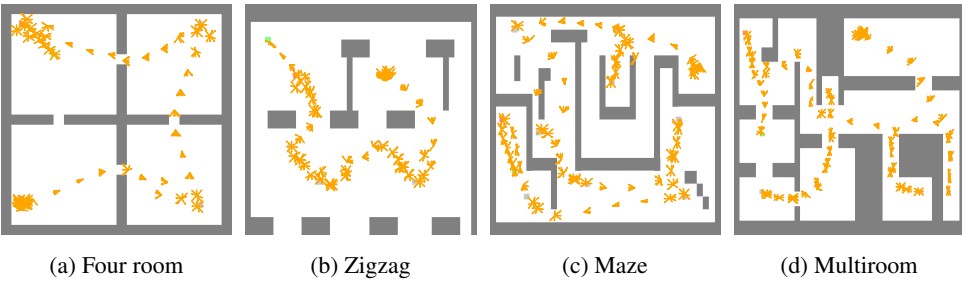

(a) Four room      (b) Zigzag      (c) Maze      (d) Multiroom

Figure 6: Visualization of IDM policies when queried for representative states and possible future states in each cardinal and diagonal direction for all four tasks. Predicted actions spread out in states where the dataset exhibits large $\Delta(s)$.

sample efficiency gains of PIDM over BC. What effect do these states have on the learned IDM policy, and how might they lead to improved efficiency?

To answer this question, we qualitatively analyze the learned PIDM policies in each task, and compute the EPE gaps for any particular state $s_t = s$ within our datasets:

$$\Delta(s) \triangleq \mathrm{Var}_{\boldsymbol{s}_{t+k}|s}\left(\mathbb{E}\left[\boldsymbol{a}_t \mid s, \boldsymbol{s}_{t+k}\right]\right), \text{ such that: } \Delta = \mathbb{E}_{\boldsymbol{s}_t}[\Delta(\boldsymbol{s}_t)]. \tag{17}$$

To approximate $\Delta(s)$ for continuous states in our 2D environment, we discretize the map with $K$-means clustering over states and then compute the sample variance over actions grouped by centroid and future states within each dataset. Figure 5 visualizes the estimated values of $\Delta(s)$ for each dataset, with 500 clusters being computed to group states. Interestingly, we observe that the human movement exhibits significantly larger action variability in states surrounding the goals which the player has to navigate to.

To qualitatively analyze the PIDM policies, we train them with all available training demonstrations. We obtain representative states by taking the centroids of $K$-means clustering (using $K = 75$ for maze and multiroom and $K = 50$ for four room and zigzag) and computing eight possible future states that are reachable within $K = 1$ step into each cardinal or diagonal direction. Then, we condition the PIDM policy with the 8 possible futures per centroid. Figure 6 visualizes the actions predicted by the PIDM policy for each centroid and future state pair. We can clearly see that the actions from the same centroid state are pointing in various directions only whenever the centroid state is close to a goal or other states with large $\Delta(s)$, as visualized in

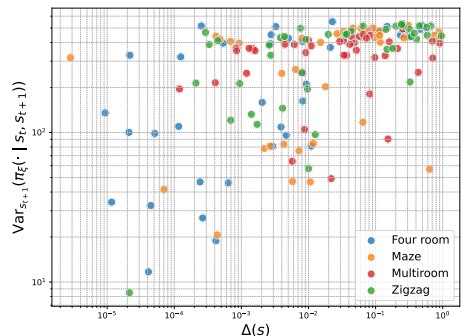

Figure 7: Correlation of state-wise action variance $\Delta(s)$ and the variance of PIDM policies.

Figure 5. This result shows that the PIDM policy learns to attend to the future state in states where the

future state helps to reduce uncertainty over the action prediction, which is precisely where Theorem 1 predicts a potential performance gain for PIDM. In contrast, in states where the action variability within the dataset is minimal, the PIDM policy exhibits significantly less diversity in predicted actions as seen by states in which most arrows point in a similar direction.

Figure 7 further connects our newly gained understanding to Theorem 1 by showing the correlation between state-wise action variability $\Delta(s)$ and the variance of the PIDM policy for varying future states, indicating that the PIDM policy exhibits higher variance for states with higher $\Delta(s)$, meaning PIDM has learned to model as predicted by our theory. The variance of the IDM policy is computed for representative centroid states over eight future states.

### 5.5 SAMPLE EFFICIENCY GAINS IN A 3D WORLD

After building an intuition for the efficiency gains of PIDM over BC both from a theoretical perspective, and under general conditions in a simplified environment, we now demonstrate similar benefits in a complex task that is representative of real-world applications. We consider the complex task that we name "Tour" as described in Section 5.1 in which the agent needs to navigate from images in the 3D world of a modern video game that requires real-time inference, with stochastic transitions, and where success is defined by achieving 11 milestones (see also Appendix C.2).

To compare agent performance on this task, BC and PIDM are trained using 5, 15, 20, 25 and 30 demonstrations. Our performance metric is the percentage of milestones that have been reached. We train BC and PIDM for 5 random seeds, and evaluate the latest checkpoint of each seed with 10 rollouts, giving a total of 50 values of the performance metric per number of demonstrations for each algorithm. Figure 1b shows PIDM achieves 95% success (on average) at the end of training, and a success rate of 87% with 15 demonstrations, while BC requires 25 demonstrations to reach a 81% success rate, so BC requires $\eta_{\text{PIDM}}(80\%) = 1.66$ times more samples than BC to reach a success rate of 80%. This confirms the potential of PIDM to improve sample efficiency over BC even in the small-data regime. Moreover, when additional data sources are available, we expect these efficiency gains to increase.

## 6 CONCLUSION

This work analyzes the performance advantages of predictive inverse dynamics models (PIDM) as an alternative to behavior cloning (BC) for offline imitation learning, particularly in low-data regimes. Through theoretical analysis and empirical experiments, we shed light onto the advantages of PIDM observed in prior studies: PIDM introduces a bias-variance tradeoff, reducing action prediction error and increasing sample efficiency by conditioning on future states—especially in regions of high uncertainty—at the cost of bias from an approximate state predictor. Moreover, we establish conditions on the state predictor bias under which PIDM is guaranteed to outperform BC. Finally, We formally motivate the use of additional data sources when available. Empirical results across navigation tasks in 2D and 3D environments confirmed sample efficiency gains, with BC requiring up to $5\times$ more demonstrations than PIDM to achieve comparable performance. Interestingly, qualitative analysis showed that learned PIDM policies attend to future states only when they provide informative context for reducing prediction variance. Altogether, this work provides a principled explanation for PIDM's effectiveness and offers insights that pave the way for more efficient imitation learning methods that leverage state prediction and future conditioning.

Although we focused on point estimators of the policy distribution for both BC and PIDM, which is a fair comparison, previous studies showed that PIDM outperforms BC with richer policy classes, like a diffusion model (Xie et al., 2025) or a transformer (Tian et al., 2025), even when they use the same dataset. Those result suggest the bias-variance we have unveiled is a feature of the PIDM architecture, independent on the modeling choices. Moreover, although the conditions that guarantee PIDM being more sample-efficient than BC have been derived for large enough samples, we have provided empirical evidence of sample efficiency gains hold and that they are affected by the state predictor bias even in the small-data regime (see Appendix A.6 and Appendix D.3).

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

# A PROOFS

## A.1 PROOF OF THEOREM 1

**Theorem 1.** $\Delta = \mathbb{E}_{\boldsymbol{s}_t}\left[\mathrm{Var}_{\boldsymbol{s}_{t+k}|\boldsymbol{s}_t}\left(\mathbb{E}\left[\boldsymbol{a}_t \mid \boldsymbol{s}_t, \boldsymbol{s}_{t+k}\right]\right)\right] \geq 0$.

*Proof:* The prediction error for these estimators is given by:

$$\mathrm{EPE}(\overline{\mu}) = \mathbb{E}_{\boldsymbol{s}_t, \boldsymbol{a}_t}\left[\left(\boldsymbol{a}_t - \overline{\mu}(\boldsymbol{s}_t)\right)^2\right], \tag{18}$$

$$\mathrm{EPE}(\overline{\xi}) = \mathbb{E}_{\boldsymbol{s}_t, \boldsymbol{a}_t, \boldsymbol{s}_{t+k}}\left[\left(\boldsymbol{a}_t - \overline{\xi}(\boldsymbol{s}_t, \boldsymbol{s}_{t+k})\right)^2\right]. \tag{19}$$

We can rewrite the EPE by using iterated expectation and replacing the definitions of optimal estimators:

$$\mathrm{EPE}(\overline{\mu}) = \mathbb{E}_{\boldsymbol{s}_t}\left[\mathbb{E}_{\boldsymbol{a}_t|\boldsymbol{s}_t}\left[\left(\boldsymbol{a}_t - \mathbb{E}\left[\boldsymbol{a}_t \mid \boldsymbol{s}_t\right]\right)^2\right]\right]$$
$$= \mathbb{E}_{\boldsymbol{s}_t}\left[\mathrm{Var}(\boldsymbol{a}_t \mid \boldsymbol{s}_t)\right] \tag{20}$$

$$\mathrm{EPE}(\overline{\xi}) = \mathbb{E}_{\boldsymbol{s}_t, \boldsymbol{s}_{t+k}}\left[\mathbb{E}_{\boldsymbol{a}_t|(\boldsymbol{s}_t, \boldsymbol{s}_{t+k})}\left[\left(\boldsymbol{a}_t - \mathbb{E}[\boldsymbol{a}_t \mid \boldsymbol{s}_t, \boldsymbol{s}_{t+k}]\right)^2\right]\right],$$
$$= \mathbb{E}_{\boldsymbol{s}_t, \boldsymbol{s}_{t+k}}\left[\mathrm{Var}(\boldsymbol{a}_t \mid \boldsymbol{s}_t, \boldsymbol{s}_{t+k})\right]. \tag{21}$$

We can further simplify Equation (20). First, we apply the law of total variance to $\text{Var}(\boldsymbol{a}_t \mid \boldsymbol{s}_t)$:

$$\text{Var}(\boldsymbol{a}_t \mid \boldsymbol{s}_t) = \mathbb{E}_{\boldsymbol{s}_{t+k} \mid \boldsymbol{s}_t} \left[ \text{Var}(\boldsymbol{a}_t \mid \boldsymbol{s}_t, \boldsymbol{s}_{t+k}) \right] + \text{Var}_{\boldsymbol{s}_{t+k} \mid \boldsymbol{s}_t} \left( \mathbb{E}[\boldsymbol{a}_t \mid \boldsymbol{s}_t, \boldsymbol{s}_{t+k}] \right). \tag{22}$$

Second, we take the expectation over $\boldsymbol{s}_t$:

$$\mathbb{E}_{\boldsymbol{s}_t}[\text{Var}(\boldsymbol{a}_t \mid \boldsymbol{s}_t)] = \mathbb{E}_{\boldsymbol{s}_t} \left[ \mathbb{E}_{\boldsymbol{s}_{t+k} \mid \boldsymbol{s}_t} \left[ \text{Var}(\boldsymbol{a}_t \mid \boldsymbol{s}_t, \boldsymbol{s}_{t+k}) \right] \right] + \mathbb{E}_{\boldsymbol{s}_t} \left[ \text{Var}_{\boldsymbol{s}_{t+k} \mid \boldsymbol{s}_t} \left( \mathbb{E}[\boldsymbol{a}_t \mid \boldsymbol{s}_t, \boldsymbol{s}_{t+k}] \right) \right]. \tag{23}$$

Third, we simplify the first term of the r.h.s.:

$$\mathbb{E}_{\boldsymbol{s}_t} \left[ \mathbb{E}_{\boldsymbol{s}_{t+k} \mid \boldsymbol{s}_t} \left[ \text{Var}(\boldsymbol{a}_t \mid \boldsymbol{s}_t, \boldsymbol{s}_{t+k}) \right] \right] = \mathbb{E}_{\boldsymbol{s}_t, \boldsymbol{s}_{t+k}}[\text{Var}(\boldsymbol{a}_t \mid \boldsymbol{s}_t, \boldsymbol{s}_{t+k})]. \tag{24}$$

Finally, we have:

$$\mathbb{E}_{\boldsymbol{s}_t}[\text{Var}(\boldsymbol{a}_t \mid \boldsymbol{s}_t)] = \mathbb{E}_{\boldsymbol{s}_t, \boldsymbol{s}_{t+k}}[\text{Var}(\boldsymbol{a} \mid \boldsymbol{s}_t, \boldsymbol{s}_{t+k})] + \mathbb{E}_{\boldsymbol{s}_t} \left[ \text{Var}_{\boldsymbol{s}_{t+k} \mid \boldsymbol{s}_t}(\mathbb{E}[\boldsymbol{a}_t \mid \boldsymbol{s}_t, \boldsymbol{s}_{t+k}]) \right]. \tag{25}$$

Now, we can easily compute the performance gap between the MSEs of both estimators:

$$\begin{aligned} \text{EPE}(\overline{\mu}) - \text{EPE}(\overline{\xi}) &= \mathbb{E}_{\boldsymbol{s}_t}[\text{Var}(\boldsymbol{a}_t \mid \boldsymbol{s}_t)] - \mathbb{E}_{\boldsymbol{s}_t, \boldsymbol{s}_{t+k}}[\text{Var}(\boldsymbol{a}_t \mid \boldsymbol{s}_t, \boldsymbol{s}_{t+k})] \\ &= \mathbb{E}_{\boldsymbol{s}_t} \left[ \text{Var}_{\boldsymbol{s}_{t+k} \mid \boldsymbol{s}_t}(\mathbb{E}[\boldsymbol{a}_t \mid \boldsymbol{s}_t, \boldsymbol{s}_{t+k}]) \right]. \end{aligned} \tag{26}$$

∎

## A.2 Proof of Corollary 1

**Corollary 1.** *Let* $\widehat{\mu}$ *and* $\widehat{\overline{\xi}}_{\widehat{p}}$ *be the estimator of the BC and IDM policies obtained with* $\mathcal{D}_n$ *and* $\mathcal{D}_{\widehat{p}, m}$, *respectively. Let the difference in the estimators' own variance and bias be given by:*

$$\delta \triangleq \mathbb{E}_{\boldsymbol{s}_t} \left[ \text{Var}(\widehat{\mu}(\boldsymbol{s}_t))] \right] - \mathbb{E}_{\boldsymbol{s}_t, \boldsymbol{s}_{t+k}} \left[ \text{Var}\left( \widehat{\overline{\xi}}_{\widehat{p}}(\boldsymbol{s}_t, \boldsymbol{s}_{t+k}) \right) \right], \tag{27}$$

$$\beta \triangleq b_{\mu}^2(\widehat{\mu}) - b_{\xi}^2(\widehat{\overline{\xi}}_{\widehat{p}}). \tag{28}$$

*And let* $\Delta$ *be given by* (6). *Then, the predictor error gap is given by:*

$$\widehat{\Delta} \triangleq \text{EPE}(\widehat{\mu}) - \text{EPE}\left( \widehat{\overline{\xi}} \right) = \Delta + \delta + \beta. \tag{29}$$

*Proof:* The EPE can be expressed as the sum of the irreducible variance and the estimator's own variance and bias. We do the derivation here for completeness:

$$\begin{aligned} \text{EPE}(\widehat{\mu}) &= \mathbb{E}_{\boldsymbol{s}_t, \boldsymbol{a}_t, \mathcal{D}_n} \left[ (\boldsymbol{a}_t - \widehat{\mu}(\boldsymbol{s}_t))^2 \right] \\ &= \mathbb{E}_{\boldsymbol{s}_t, \boldsymbol{a}_t, \mathcal{D}_n} \left[ (\boldsymbol{a}_t - \widehat{\mu}(\boldsymbol{s}_t) + \overline{\mu}(\boldsymbol{s}_t) - \overline{\mu}(\boldsymbol{s}_t))^2 \right] \\ &= \mathbb{E}_{\boldsymbol{s}_t, \boldsymbol{a}_t, \mathcal{D}_n} \left[ ((\boldsymbol{a}_t - \overline{\mu}(\boldsymbol{s}_t)) + (\overline{\mu}(\boldsymbol{s}_t) - \widehat{\mu}(\boldsymbol{s}_t)))^2 \right] \\ &= \mathbb{E}_{\boldsymbol{s}_t, \boldsymbol{a}_t} \left[ (\boldsymbol{a}_t - \overline{\mu}(\boldsymbol{s}_t))^2 \right] + \mathbb{E}_{\boldsymbol{s}_t, \mathcal{D}_n} \left[ (\overline{\mu}(\boldsymbol{s}_t) - \widehat{\mu}(\boldsymbol{s}_t))^2 \right] \\ &\quad + 2\mathbb{E}_{\boldsymbol{s}_t, \boldsymbol{a}_t, \mathcal{D}_n} \left[ (\boldsymbol{a}_t - \overline{\mu}(\boldsymbol{s}_t))(\overline{\mu}(\boldsymbol{s}_t) - \widehat{\mu}(\boldsymbol{s}_t)) \right]. \end{aligned} \tag{30}$$

The first term is the expected conditional variance:

$$\begin{aligned} \mathbb{E}_{\boldsymbol{s}_t, \boldsymbol{a}_t} \left[ (\boldsymbol{a}_t - \overline{\mu}(\boldsymbol{s}_t))^2 \right] &= \mathbb{E}_{\boldsymbol{s}_t} \left[ \mathbb{E}_{\boldsymbol{a}_t \mid \boldsymbol{s}_t} \left[ (\boldsymbol{a}_t - \mathbb{E}[\boldsymbol{a}_t \mid \boldsymbol{s}_t])^2 \right] \right] \\ &= \mathbb{E}_{\boldsymbol{s}_t} \left[ \text{Var}(\boldsymbol{a}_t \mid \boldsymbol{s}_t) \right]. \end{aligned} \tag{31}$$

The cross-term vanishes:

$$\begin{aligned} \mathbb{E}_{\boldsymbol{s}_t, \boldsymbol{a}_t} \left[ (\boldsymbol{a}_t - \overline{\mu}(\boldsymbol{s}_t))(\overline{\mu}(\boldsymbol{s}_t) - \widehat{\mu}(\boldsymbol{s}_t)) \right] &= \mathbb{E}_{\boldsymbol{s}_t} \left[ \mathbb{E}_{\boldsymbol{a}_t \mid \boldsymbol{s}_t} \left[ \boldsymbol{a}_t - \overline{\mu}(\boldsymbol{s}_t) \right] (\overline{\mu}(\boldsymbol{s}_t) - \widehat{\mu}(\boldsymbol{s}_t)) \right] \\ &= \mathbb{E}_{\boldsymbol{s}_t} \left[ (\mathbb{E}[\boldsymbol{a}_t \mid \boldsymbol{s}_t] - \overline{\mu}(\boldsymbol{s}_t))(\overline{\mu}(\boldsymbol{s}_t) - \widehat{\mu}(\boldsymbol{s}_t)) \right] \\ &= 0. \end{aligned} \tag{32}$$

The second term decomposes in the expected variance and expected bias terms:

$$\begin{aligned} \mathbb{E}_{\boldsymbol{s}_t, \mathcal{D}_n} \left[ (\overline{\mu}(\boldsymbol{s}_t) - \widehat{\mu}(\boldsymbol{s}_t))^2 \right] &= \mathbb{E}_{\boldsymbol{s}_t} \left[ \mathbb{E}_{\mathcal{D}_n} \left[ (\mathbb{E}_{\mathcal{D}_n}[\widehat{\mu}(\boldsymbol{s}_t)] - \widehat{\mu}(\boldsymbol{s}_t) + \overline{\mu}(\boldsymbol{s}_t) - \mathbb{E}_{\mathcal{D}_n}[\widehat{\mu}(\boldsymbol{s}_t)])^2 \right] \right] \\ &= \mathbb{E}_{\boldsymbol{s}_t} \left[ \mathbb{E}_{\mathcal{D}_n} \left[ (\mathbb{E}_{\mathcal{D}_n}[\widehat{\mu}(\boldsymbol{s}_t)] - \widehat{\mu}(\boldsymbol{s}_t))^2 \right] \right] + \mathbb{E}_{\boldsymbol{s}_t} \left[ \mathbb{E}_{\mathcal{D}_n} \left[ (\overline{\mu}(\boldsymbol{s}_t) - \mathbb{E}_{\mathcal{D}_n}[\widehat{\mu}(\boldsymbol{s}_t)])^2 \right] \right] \\ &\quad + 2\mathbb{E}_{\boldsymbol{s}_t} \left[ \mathbb{E}_{\mathcal{D}_n}[(\mathbb{E}_{\mathcal{D}_n}[\widehat{\mu}(\boldsymbol{s}_t)] - \widehat{\mu}(\boldsymbol{s}_t))(\overline{\mu}(\boldsymbol{s}_t) - \mathbb{E}_{\mathcal{D}_n}[\widehat{\mu}(\boldsymbol{s}_t)])] \right] \\ &= \mathbb{E}_{\boldsymbol{s}_t} \left[ \text{Var}(\widehat{\mu}(\boldsymbol{s}_t)) \right] + \mathbb{E}_{\boldsymbol{s}_t} \left[ (\overline{\mu}(\boldsymbol{s}_t) - \mathbb{E}_{\mathcal{D}_n}[\widehat{\mu}(\boldsymbol{s}_t)])^2 \right], \end{aligned} \tag{33}$$

where the cross-term also vanished. Putting the terms together, we have:

$$\text{EPE}(\widehat{\mu}) = \mathbb{E}_{\boldsymbol{s}_t}\left[\text{Var}(\boldsymbol{a}_t \mid \boldsymbol{s}_t)\right] + \mathbb{E}_{\boldsymbol{s}_t}\left[\text{Var}\left(\widehat{\mu}\left(\boldsymbol{s}_t\right)\right)\right] + \mathbb{E}_{\boldsymbol{s}_t}\left[\left(\overline{\mu}(\boldsymbol{s}_t) - \mathbb{E}_{\mathcal{D}_n}\left[\widehat{\mu}(\boldsymbol{s}_t)\right]\right)^2\right]. \tag{34}$$

Following the same approach for $\widehat{\xi}_{\widehat{p}}$, we have:

$$\text{EPE}(\widehat{\xi}_{\widehat{p}}) = \mathbb{E}_{\boldsymbol{s}_t, \boldsymbol{s}_{t+k}}\left[\text{Var}(\boldsymbol{a}_t \mid \boldsymbol{s}_t, \boldsymbol{s}_{t+k})\right] + \mathbb{E}_{\boldsymbol{s}_t, \boldsymbol{s}_{t+k}}\left[\text{Var}\left(\widehat{\xi}_{\widehat{p}}\left(\boldsymbol{s}_t, \boldsymbol{s}_{t+k}\right)\right)\right]$$
$$+ \mathbb{E}_{\boldsymbol{s}_t, \boldsymbol{s}_{t+k}}\left[\left(\overline{\xi}(\boldsymbol{s}_t, \boldsymbol{s}_{t+k}) - \mathbb{E}_{\mathcal{D}_{\widehat{p}, m}}\left[\widehat{\xi}_{\widehat{p}}(\boldsymbol{s}_t, \boldsymbol{s}_{t+k})\right]\right)^2\right]. \tag{35}$$

Subtracting (35) from (34) and grouping terms according to (6) and (27)–(28) concludes the proof. ∎

### A.3 PROOF OF THEOREM 2

We need the information inequality for any estimator, which is given by the following standard result.

**Lemma 1.** *Consider a distribution $f_\theta(\cdot)$ with parameter $\theta$ and Fisher information $F_\theta$. The MSE of any estimator $\widehat{\theta}$ of $\theta$, obtained from $n$ samples drawn i.i.d. from $f_\theta(\cdot)$, satisfies this information inequality:*

$$\mathbb{E}\left[\left(\widehat{\theta} - \theta\right)^2\right] \geq \frac{\left(\frac{\partial}{\partial\theta}b_\theta(\widehat{\theta}) + 1\right)^2}{nF_\theta} + b_\theta^2(\widehat{\theta}). \tag{36}$$

*Proof:* See, e.g., Cover (2005, Chapter 11) and combine Equation (11.290), which states the information inequality for any estimator for a single sample, with Equation (11.279), which defines the Fisher information for $n$ i.i.d. random variables. ∎

We are now ready to proof Theorem 2.

**Theorem 2.** *Let $\widehat{\mu}_n$ and $\widehat{\xi}_{\widehat{p}, m}$ be asymptotically efficient estimator of the BC and IDM policies obtained with $\mathcal{D}_n$ and $\mathcal{D}_{\widehat{p}, m}$, respectively, where $n$ and $m$ denote the minimum number of samples required to achieve error level $\varepsilon$. Let $F_\mu$ and $F_\xi$ exist, and let $\pi_\xi$ satisfy regularity conditions (for differentiating under the integral sign.) Then, for large enough $n$ and $m$, we have:*

$$\eta \triangleq \frac{n}{m} \approx \frac{F_\xi}{F_\mu} \frac{\left(\frac{\partial}{\partial\mu}b_\mu(\widehat{\mu}_n) + 1\right)^2}{\left(\frac{\partial}{\partial\xi}b_\xi(\widehat{\xi}_{\widehat{p}, m}) + 1\right)^2} \left(1 + \frac{\Delta + b_\mu^2(\widehat{\mu}_n) - b_\xi^2(\widehat{\xi}_{\widehat{p}, m})}{\varepsilon - \mathbb{E}_{\boldsymbol{s}_t}\left[\text{Var}(\boldsymbol{a}_t \mid \boldsymbol{s}_t)\right] - b_\mu^2(\widehat{\mu}_n)}\right). \tag{37}$$

*Proof:* Asymptotic efficiency means that for large enough number of samples, the MSE approximately meets the lower bound in Lemma 1 with equality. Hence, from (34) and (36), we have:

$$\text{EPE}(\widehat{\mu}_n) = \mathbb{E}_{\boldsymbol{s}_t}\left[\text{Var}(\boldsymbol{a}_t \mid \boldsymbol{s}_t)\right] + \mathbb{E}_{\boldsymbol{s}_t}\left[\text{Var}(\widehat{\mu}_n(\boldsymbol{s}_t))\right] + b_\mu^2(\widehat{\mu}_n)$$
$$\approx \mathbb{E}_{\boldsymbol{s}_t}\left[\text{Var}(\boldsymbol{a}_t \mid \boldsymbol{s}_t)\right] + \frac{\left(\frac{\partial}{\partial\mu}b_\mu(\widehat{\mu}_n) + 1\right)^2}{nF_\mu} + b_\mu^2(\widehat{\mu}_n). \tag{38}$$

Since $\text{EPE}(\widehat{\mu}_n) = \varepsilon$, we can solve for $n$:

$$n \approx \frac{\left(\frac{\partial}{\partial\mu}b_\mu(\widehat{\mu}_n) + 1\right)^2}{F_\mu\left(\varepsilon - \mathbb{E}_{\boldsymbol{s}_t}\left[\text{Var}(\boldsymbol{a}_t \mid \boldsymbol{s}_t)\right] - b_\mu^2(\widehat{\mu}_n)\right)}. \tag{39}$$

Following the same reasoning for $\text{EPE}(\widehat{\xi}_{\widehat{p}, m})$, we get:

$$m \approx \frac{\left(\frac{\partial}{\partial\xi}b_\xi(\widehat{\xi}_{\widehat{p}, m}) + 1\right)^2}{F_\xi\left(\varepsilon - \mathbb{E}_{\boldsymbol{s}_t, \boldsymbol{s}_{t+k}}\left[\text{Var}(\boldsymbol{a}_t \mid \boldsymbol{s}_t, \boldsymbol{s}_{t+k})\right] - b_\xi^2(\widehat{\xi}_{\widehat{p}, m})\right)}. \tag{40}$$

Note that (slightly abused notation) the bias derivative terms are given by:

$$\frac{\partial}{\partial \mu} b_\mu(\widehat{\mu}_n) \triangleq \frac{\partial}{\partial \mu} \mathbb{E}_{\boldsymbol{s}_t} \left[ E_{\mathcal{D}_n} \left[ \widehat{\mu}_n(\boldsymbol{s}_t) \right] - \overline{\mu}(\boldsymbol{s}_t) \right], \tag{41}$$

$$\frac{\partial}{\partial \xi} b_\xi(\widehat{\xi}_{\widehat{p},m}) \triangleq \frac{\partial}{\partial \xi} \mathbb{E}_{\boldsymbol{s}_t, \boldsymbol{s}_{t+k}} \left[ \mathbb{E}_{\mathcal{D}_{\widehat{p},m}} \left[ \widehat{\xi}_{\widehat{p},m}(\boldsymbol{s}_t, \boldsymbol{s}_{t+k}) \right] - \overline{\xi}(\boldsymbol{s}_t, \boldsymbol{s}_{t+k}) \right]. \tag{42}$$

Using the definition of the sample efficiency ratio, we have:

$$\eta \triangleq \frac{n}{m}$$

$$\approx \frac{F_\xi}{F_\mu} \frac{\left( \frac{\partial}{\partial \mu} b_\mu(\widehat{\mu}_n) + 1 \right)^2}{\left( \frac{\partial}{\partial \xi} b_\xi(\widehat{\xi}_{\widehat{p},m}) + 1 \right)^2} \left( \frac{\varepsilon - \mathbb{E}_{\boldsymbol{s}_t, \boldsymbol{s}_{t+k}} \left[ \mathrm{Var}(\boldsymbol{a}_t \mid \boldsymbol{s}_t, \boldsymbol{s}_{t+k}) \right] - b_\xi^2(\widehat{\xi}_{\widehat{p},m})}{\varepsilon - \mathbb{E}_{\boldsymbol{s}_t} \left[ \mathrm{Var}(\boldsymbol{a}_t \mid \boldsymbol{s}_t) \right] - b_\mu^2(\widehat{\mu}_n)} \right). \tag{43}$$

From the first line of (26), we obtain this identity:

$$\mathbb{E}_{\boldsymbol{s}_t, \boldsymbol{s}_{t+k}} \left[ \mathrm{Var}(\boldsymbol{a}_t \mid \boldsymbol{s}_t, \boldsymbol{s}_{t+k}) \right] = \mathbb{E}_{\boldsymbol{s}_t} \left[ \mathrm{Var}(\boldsymbol{a}_t \mid \boldsymbol{s}_t) \right] - \Delta. \tag{44}$$

Expanding (44) in (43) and adding and subtracting $b_\mu^2(\widehat{\mu}_n)$ to the numerator yields:

$$\eta \approx \frac{F_\xi}{F_\mu} \frac{\left( \frac{\partial}{\partial \mu} b_\mu(\widehat{\mu}_n) + 1 \right)^2}{\left( \frac{\partial}{\partial \xi} b_\xi(\widehat{\xi}_{\widehat{p},m}) + 1 \right)^2} \left( \frac{\varepsilon - \mathbb{E}_{\boldsymbol{s}_t} \left[ \mathrm{Var}(\boldsymbol{a}_t \mid \boldsymbol{s}_t) \right] - b_\mu^2(\widehat{\mu}_n) + b_\mu^2(\widehat{\mu}_n) - b_\xi^2(\widehat{\xi}_{\widehat{p},m}) + \Delta}{\varepsilon - \mathbb{E}_{\boldsymbol{s}_t} \left[ \mathrm{Var}(\boldsymbol{a}_t \mid \boldsymbol{s}_t) \right] - b_\mu^2(\widehat{\mu}_n)} \right).$$

$$\tag{45}$$

Simplifying terms concludes the proof. ∎

### A.4 PROOF OF THEOREM 3

We need the following lemma that shows that the ratio of Fisher information for the estimators of the BC and IDM policies is greater than or equal to one.

**Lemma 2.** *Assume $F_\mu$ and $F_\xi$ exist. Under regularity conditions (for differentiating under the integral sign), we have: $\frac{F_\xi}{F_\mu} \geq 1$.*

*Proof:* Since BC policy can be obtained as the marginal of the IDM policy, it is convenient to write $\pi_{\mu(\xi)}$ and make explicit that the BC policy parameter is a function of the IDM policy parameter:

$$\pi_{\mu(\xi)}(a_t \mid s_t) = \int_{\mathbb{S}} p^\star(s_{t+k} \mid s_t) \pi_\xi(a_t \mid s_t, s_{t+k}) ds_{t+k}. \tag{46}$$

The Fisher information for the BC and IDM policies is given by:

$$F_\mu \triangleq \mathbb{E} \left[ \left( \frac{\partial}{\partial \xi} \ln \pi_{\mu(\xi)}(\boldsymbol{a}_t \mid \boldsymbol{s}_t) \right)^2 \right], \tag{47}$$

$$F_\xi \triangleq \mathbb{E} \left[ \left( \frac{\partial}{\partial \xi} \ln \pi_\xi(\boldsymbol{a}_t \mid \boldsymbol{s}_t, \boldsymbol{s}_{t+k}) \right)^2 \right]. \tag{48}$$

We treat the state predictor as fixed, meaning that we compute the curvature of the log-likelihood with respect to $\pi_\xi$ while holding $p^\star$ constant. Hence, we can expand the term $\frac{\partial}{\partial \xi} \ln \pi_{\mu(\xi)}(\boldsymbol{a}_t \mid \boldsymbol{s}_t)$ in

(46) as follows:

$$
\frac{\partial}{\partial \xi} \ln \pi_{\mu(\xi)}(\boldsymbol{a}_t \mid \boldsymbol{s}_t) = \frac{\frac{\partial}{\partial \xi} \pi_{\mu(\xi)}(\boldsymbol{a}_t \mid \boldsymbol{s}_t)}{\pi_{\mu(\xi)}(\boldsymbol{a}_t \mid \boldsymbol{s}_t)}
$$

$$
= \frac{\frac{\partial}{\partial \xi} \int_{\mathbb{S}} p^{\star}(s_{t+k} \mid s_t) \pi_{\xi}(a_t \mid s_t, s_{t+k}) ds_{t+k}}{\pi_{\mu(\xi)}(\boldsymbol{a}_t \mid \boldsymbol{s}_t)}
$$

$$
= \frac{\int_{\mathbb{S}} p^{\star}(s_{t+k} \mid s_t) \left( \frac{\partial}{\partial \xi} \pi_{\xi}(a_t \mid s_t, s_{t+k}) \right) ds_{t+k}}{\pi_{\mu(\xi)}(\boldsymbol{a}_t \mid \boldsymbol{s}_t)}
$$

$$
= \frac{\int_{\mathbb{S}} p^{\star}(s_{t+k} \mid s_t) \pi_{\xi}(\boldsymbol{a}_t \mid \boldsymbol{s}_t, \boldsymbol{s}_{t+k}) \left( \frac{\partial}{\partial \xi} \ln \pi_{\xi}(\boldsymbol{a}_t \mid \boldsymbol{s}_t, \boldsymbol{s}_{t+k}) \right) ds_{t+k}}{\pi_{\mu(\xi)}(\boldsymbol{a}_t \mid \boldsymbol{s}_t)}
$$

$$
= \frac{\int_{\mathbb{S}} P(s_{t+k} | s_t, a_t) \left( \frac{\partial}{\partial \xi} \ln \pi_{\xi}(\boldsymbol{a}_t \mid \boldsymbol{s}_t, \boldsymbol{s}_{t+k}) \right) ds_{t+k}}{\pi_{\mu(\xi)}(\boldsymbol{a}_t \mid \boldsymbol{s}_t)}
$$

$$
= \mathbb{E}_{\boldsymbol{s}_{t+k} | \boldsymbol{s}_t, \boldsymbol{a}_t} \left[ \frac{\partial}{\partial \xi} \ln \pi_{\xi}(\boldsymbol{a}_t \mid \boldsymbol{s}_t, \boldsymbol{s}_{t+k}) \right], \tag{49}
$$

where we commuted the partial derivative and the integral (allowed by the regularity conditions); used Bayes to obtain the following distribution:

$$
P(s_{t+k} | s_t, a_t) \triangleq \frac{p^{\star}(s_{t+k} | s_t) \pi_{\xi}(a_t | s_t, s_{t+k})}{\pi_{\mu(\xi)}(a_t | s_t)}; \tag{50}
$$

and used this identity:

$$
\frac{\partial}{\partial \xi} \pi_{\xi}(\boldsymbol{a}_t \mid \boldsymbol{s}_t, \boldsymbol{s}_{t+k}) = \pi_{\xi}(\boldsymbol{a}_t \mid \boldsymbol{s}_t, \boldsymbol{s}_{t+k}) \frac{\partial}{\partial \xi} \ln \pi_{\xi}(\boldsymbol{a}_t \mid \boldsymbol{s}_t, \boldsymbol{s}_{t+k}). \tag{51}
$$

In summary:

$$
\frac{\partial}{\partial \xi} \ln \pi_{\mu(\xi)}(\boldsymbol{a}_t \mid \boldsymbol{s}_t) = \mathbb{E}_{\boldsymbol{s}_{t+k} | \boldsymbol{s}_t, \boldsymbol{a}_t} \left[ \frac{\partial}{\partial \xi} \ln \pi_{\xi}(\boldsymbol{a}_t \mid \boldsymbol{s}_t, \boldsymbol{s}_{t+k}) \right]. \tag{52}
$$

By Jensen's inequality, we have:

$$
\left( \frac{\partial}{\partial \xi} \ln \pi_{\mu(\xi)}(\boldsymbol{a}_t \mid \boldsymbol{s}_t) \right)^2 = \mathbb{E}_{\boldsymbol{s}_{t+k} | \boldsymbol{s}_t, \boldsymbol{a}_t} \left[ \frac{\partial}{\partial \xi} \ln \pi_{\xi}(\boldsymbol{a}_t \mid \boldsymbol{s}_t, \boldsymbol{s}_{t+k}) \right]^2
$$

$$
\leq \mathbb{E}_{\boldsymbol{s}_{t+k} | \boldsymbol{s}_t, \boldsymbol{a}_t} \left[ \left( \frac{\partial}{\partial \xi} \ln \pi_{\xi}(\boldsymbol{a}_t \mid \boldsymbol{s}_t, \boldsymbol{s}_{t+k}) \right)^2 \right]. \tag{53}
$$

Since the inequality in (53) holds pointwise, taking expectation on both sides keeps the direction of the inequality:

$$
F_{\mu} = \int p(s_t) \pi_{\mu(\xi)}(a_t | s_t) \left( \frac{\partial}{\partial \xi} \ln \pi_{\mu(\xi)}(\boldsymbol{a}_t \mid \boldsymbol{s}_t) \right)^2 da_t ds_t
$$

$$
\leq \int p(s_t) \pi_{\mu(\xi)}(a_t | s_t) P(s_{t+k} | s_t, a_t) \left[ \left( \frac{\partial}{\partial \xi} \ln \pi_{\xi}(a_t \mid s_t, s_{t+k}) \right)^2 \right] ds_{t+k} da_t ds_t
$$

$$
= \int p(s_t) \pi_{\mu(\xi)}(a_t | s_t) \frac{p^{\star}(s_{t+k} | s_t) \pi_{\xi}(a_t | s_t, s_{t+k})}{\pi_{\mu(\xi)}(a_t | s_t)} \left[ \left( \frac{\partial}{\partial \xi} \ln \pi_{\xi}(a_t \mid s_t, s_{t+k}) \right)^2 \right] ds_{t+k} da_t ds_t
$$

$$
= \int p(s_t) p^{\star}(s_{t+k} | s_t) \pi_{\xi}(a_t | s_t, s_{t+k}) \left[ \left( \frac{\partial}{\partial \xi} \ln \pi_{\xi}(a_t \mid s_t, s_{t+k}) \right)^2 \right] ds_{t+k} da_t ds_t
$$

$$
= F_{\xi}. \tag{54}
$$

We conclude that $F_{\xi} \geq F_{\mu}$, or equivalently: $F_{\xi}/F_{\mu} \geq 1$. ∎

We are ready to prove Theorem 3.

**Theorem 3.** *Under the conditions of Theorem 2, assume the following condition holds:*

$$\overline{\varepsilon} + \Delta \geq b_\xi^2(\widehat{\xi}_{\widehat{p},m}) + \left(\overline{\varepsilon} - b_\mu^2(\widehat{\mu}_n)\right) \frac{\left(\frac{\partial}{\partial \xi} b_\xi(\widehat{\xi}_{\widehat{p},m}) + 1\right)^2}{\left(\frac{\partial}{\partial \mu} b_\mu(\widehat{\mu}_n) + 1\right)^2}, \tag{55}$$

*where $\overline{\varepsilon} \triangleq \varepsilon - \mathbb{E}_{\boldsymbol{s}_t}\left[\mathrm{Var}(\boldsymbol{a}_t \mid \boldsymbol{s}_t)\right]$. Then: $\eta \gtrsim 1$.*

*Proof:* We need to find the conditions on the bias term $b_\xi$ that make (37) greater than or equal to one:

$$\frac{F_\xi}{F_\mu} \frac{\left(\frac{\partial}{\partial \mu} b_\mu(\widehat{\mu}_n) + 1\right)^2}{\left(\frac{\partial}{\partial \xi} b_\xi(\widehat{\xi}_{\widehat{p},m}) + 1\right)^2} \left(1 + \frac{\Delta + b_\mu^2(\widehat{\mu}_n) - b_\xi^2(\widehat{\xi}_{\widehat{p},m})}{\varepsilon - \mathbb{E}_{\boldsymbol{s}_t}\left[\mathrm{Var}(\boldsymbol{a}_t \mid \boldsymbol{s}_t)\right] - b_\mu^2(\widehat{\mu}_n)}\right) \geq 1. \tag{56}$$

Lemma 2 ensures that $\frac{F_\xi}{F_\mu} \geq 1$, so it doesn't affect the inequality. Hence, we just have to rearrange terms to obtain (55). ∎

### A.5 PROOF OF COROLLARY 2

**Corollary 2.** *Under the conditions of Theorem 2, if $\widehat{\xi}_{\widehat{p},m}$ is asymptotically unbiased, then $\eta \gtrsim 1$.*

*Proof:* The terms dependent on $b_\mu$ can only reduce the gap in the inequality (by subtracting from and scaling down the contribution of $b_\xi$). Hence, making $b_\mu = 0$ ensures a more conservative condition, which only depends on $b_\xi$:

$$b_\xi^2(\widehat{\xi}_{\widehat{p},m}) + \overline{\varepsilon} \left(\frac{\partial}{\partial \xi} b_\xi(\widehat{\xi}_{\widehat{p},m}) + 1\right)^2 \leq \overline{\varepsilon} + \Delta. \tag{57}$$

Since the estimators are asymptotically unbiased, for large enough $n$ and $m$, we have: $b_\xi(\widehat{\xi}) \approx 0$. Using this approximation in (57) reduces the inequality to: $\Delta \geq 0$, which is always guaranteed, as stated by Theorem 1. ∎

### A.6 DISCUSSION ON FINITE-TIME PERFORMANCE

The asymptotic covariance often predicts finite-time performance well, as the Central Limit Theorem approximation is accurate for moderately large number of samples. For instance, the Berry–Esseen theorem states that, for i.i.d. samples, the convergence rate towards asymptotic normality is $1/\sqrt{n}$, where $n$ is the number of samples.

How this impacts our results is most clearly seen in the asymptotically unbiased case discussed in Corollary 2, where $\eta \geq 1$ holds because $\frac{F_\xi}{F_\mu} \geq 1$ (see Lemma 2). Since the Fisher information is the inverse of the asymptotic covariance, this ratio implies that the IDM estimators' asymptotic covariance is no larger than that of BC. This is a fundamental fact in any data regime.

Furthermore, Corollary 1 and Theorem 2 show the same bias-variance tradeoff for EPE and sample efficiency: the variance reduction of the IDM increases the gap, while the state predictor introduces bias that reduces the gap. Combining the facts that Corollary 1 holds for any number of samples and that the fundamental mechanism is the same for both Corollary 1 and Theorem 2 suggests that Theorem 2, hence Theorem 3 and Corollary 2 should also hold more generally than in the asymptotic regime. Indeed, Section 5 and Appendix D.3 provide empirical evidence of efficiency gains and of the role of the state predictor bias even in the small-data regime.

## B DETAILS FOR 2D NAVIGATION ENVIRONMENT AND EXPERIMENTS

In this section, we describe further details about the 2D navigation environment and the experiments in this setting.

Table 2: Statistics of all four 2D navigation tasks and the human datasets. The first four columns correspond to properties of the tasks, given by the number of goals, maximum number of time steps to complete the task, and the state dimensionality, while the last four columns correspond to the total number of trajectories/ time steps within the collected dataset (across all 50 trajectories) and statistics over the trajectory length.

| Task | Num goals | Max time steps | $|s|$ | Total steps | Trajectory length | | |
|------|-----------|----------------|-------|-------------|------|------|------|
| | | | | | Min | Avg | Max |
| Four room | 4 | 200 | 14 | 5821 | 103 | 116.42 | 154 |
| Zigzag | 6 | 150 | 20 | 4009 | 66 | 80.18 | 106 |
| Maze | 10 | 300 | 32 | 9785 | 176 | 195.70 | 227 |
| Multiroom | 6 | 500 | 20 | 12 961 | 241 | 259.22 | 314 |

## B.1 ADDITIONAL ENVIRONMENT DETAILS

Tasks within the 2D navigation environment specify a layout of the environment and differ in the number of goals. The general setting stays the same with each task specifying an order to its goals and the agent needs to reach a goal before being able to reach any subsequent goals. This setup makes these tasks punishing since missing any goal will mean that subsequent goals cannot be reached anymore unless the agent returns back to the currently required goal. An episode within any task finishes after all goals have been reached, or after a maximum number of time steps has been reached. The state dimensionality, number of goals, and maximum number of time steps for each task is listed in Table 2.

In all tasks, we introduce stochasticity in the transition function through Gaussian noise. Instead of displacing the agent based on its selected action $a \in [-1, 1]^2$ alone, we displace the agent based on clipped noise-added actions:

$$\text{clip}(a + \epsilon, -1, 1) \quad \text{with} \quad \epsilon \sim \mathcal{N}(0, 0.2 \cdot \mathbb{1}) \tag{58}$$

We emphasize that the sampled noise is *not* modifying the actions but rather modeled as part of the environment, meaning that, from the perspective of the agent, the environment transitions are stochastic given a state and action. The agent will bounce off any walls that it collides with with walls being visualized as black bars in all figures.

## B.2 DATASET DETAILS

Table 2 shows statistics for each 2D navigation task and the collected human dataset. During data collection, the human player was instructed to collect high-quality trajectories that reach all goals as fast as possible. The player controlled the movement of the controllable agent using the joystick of a gamepad controller. We note that the player was unaware of the data analyses that we conducted to avoid any risk of introducing bias.

## B.3 HYPERPARAMETER SEARCH

To ensure fair comparison, we conducted a comparable hyperparameter search for both BC and PIDM in the multiroom task using 50 training demonstrations. First, we conducted a hyperparameter search over the model architecture considering sixteen different sizes of the MLP network architecture, the use of normalization in the network (either batch normalization, layer normalization, or no normalization), and learning rate with three constant candidate learning rates ($1e^{-6}, 1e^{-5}, 1e^{-4}$. The considered architectures consisted of any of five MLP blocks before any potential normalization layer and any of the five MLP blocks after the normalization. The considered network blocks were:

1. MLP(256)

2. MLP(256, 128)

3. MLP(512, 256)

4. MLP(512, 1024, 256)

5. MLP(1024, 2048, 512)

From this search, we identified a single network architecture that performed best for BC and among the best for PIDM to keep for consistent comparisons thereafter. The architecture consists of network block MLP(512, 1024, 256) followed by batch normalization before MLP(256, 2) with the last 2D layer outputting the action logits. We apply ReLU activation in between all layers and $tanh$ activation to the output logits.

After fixing the network architecture, we still found some training instability for BC and IDM so we decided to further tune the learning rate for BC and IDM by searching over 14 learning rate configurations defined by their initial learning rate, and potential learning rate scheduling, and considered each configuration with and without gradient norm clipping. We first tuned the learning rate configuration for BC and IDM in multiroom after which we found IDM training to be stable across tasks. For BC, we further tuned the learning rate for each individual task to obtain stable training results. The identified learning rates are shown in the table below.

Table 3: Learning rate configuration for each task and algorithm

| Task | BC configuration | IDM configuration |
|---|---|---|
| Four room | Linear decay $1e^{-3} \to 1e^{-6}$ over $50\,000$ steps + grad norm clipping | constant $1e^{-5}$ |
| Zigzag | Linear decay $1e^{-4} \to 1e^{-6}$ over $50\,000$ steps + grad norm clipping | constant $1e^{-5}$ |
| Maze | Linear decay $1e^{-4} \to 1e^{-6}$ over $50\,000$ steps + grad norm clipping | constant $1e^{-5}$ |
| Multiroom | Linear decay $1e^{-4} \to 1e^{-6}$ over $50\,000$ steps | constant $1e^{-5}$ |

## C DETAILS FOR COMPLEX TASK IN 3D-WORLD

### C.1 DATASET DETAILS

The dataset consists of 30 demonstrations collected by a human playing the game. Table 4 shows the number of steps and length (seconds) of the demonstrations in the dataset.

Table 4: Statistics of demonstrations of "Tour" task.

| | Total steps | | | Trajectory length (in seconds) | | |
|---|---|---|---|---|---|---|
| Task | Min | Avg | Max | Min | Avg | Max |
| Tour | 1006 | $1067.2 \pm 29.4$ | 1139 | 33.83 | $35.91 \pm 0.99$ | 38.29 |

### C.2 ADDITIONAL ENVIRONMENT DETAILS

Table 5 contains the 11 milestones required to complete the "Tour" task.

### C.3 EVALUATION PROTOCOL

Two human experts that were familiar with the task evaluated all the rollouts. The evaluation was blind to avoid cognitive bias, since the evaluators did not know whether the rollout they were evaluating corresponded to BC or PIDM. For each rollout, they checked if the agent achieved every milestone of the task, scoring with value 1 if the milestone was achieved and 0 otherwise, so the maximum score per rollout is 11 (the number of milestones). However, we report performance in terms of % of this maximum score.

Table 5: Milestones of "Tour" task in Bleeding Edge with corresponding thumbnails

| # | Milestone | Thumbnail |
|---|---|---|
| 1 | Start off with a sharp left 180° turn |  |
| 2 | Navigate towards the first health marker and grab it |  |
| 3 | Cross the main floor of the Dojo |  |
| 4 | Take a left onto the ramp |  |
| 5 | Turn while going up and stay on the ramp for 6-7 secs |  |
| 6 | Right turn and navigate the corridor |  |
| 7 | Circumvent the box by steering left |  |
| 8 | Navigate towards the second health marker and grab it |  |
| 9 | Pass through the final corridor |  |
| 10 | Hit the Gong |  |
| 11 | Stop and don't move anymore |  |

## C.4 Additional Algorithmic Details

### C.4.1 Vision encoder

We use "theia-base-patch16-224-cddsv" from Huggingface as pretrained vision encoder. The vision encoder remains frozen during training (and evaluation). Each video frame is passed to this encoder, which generates an embedding vector of length 768. This embedding vector of the current frame is the input to the BC policy. While the embedding of the current and future frames are the input to the state encoder of the PIDM.

### C.4.2 Hyperparameter Search

To ensure fair comparison and some degree of generalization, we conducted a comparable hyperparameter search for both BC and PIDM in a different more complex task, with more milestones, for which none of the algorithms could achieve 100% performance after being trained with a dataset of 30 demonstrations. We used the results from the hyperparameter search in the 2D environment as a basis, with ReLu activations in between all layers and batch normalization at the output of the state encoder. The output was a *tanh* activation. We evaluated two different sizes of the MLP network architecture, under two learning rates. The considered MLP network blocks were:

1. State encoder: MLP(1024, 512, 512), Policy: MLP(512, 256)

2. State encoder: MLP(1024, 2048, 1024, 512, 512), Policy: MLP(512, 512, 256)

We also tried two learning rates per algorithm, namely linear decay $1e\text{-}3 \rightarrow 1e\text{-}6$ and $5e\text{-}5$ for PIDM, and linear decay $1e\text{-}4 \rightarrow 1e\text{-}6$ and $1e\text{-}4$ for BC, with decay for 60,000 steps. Other hyperparameters that remained constant where: training lasted 60,000 steps, optimization algorithm was Adam with standard parameters ($\beta_1 = 0.9$, $\beta_2 = 0.999$, $\epsilon = 1e\text{-}8$), and batch size was 4096.

We observed the small network blocks with linear decay was the best combination, and BC (88%) achieved slightly higher average performance than PIDM (86%) for that task, but not statistically significant. For training in the "Tour" task, we used this configuration and used the rest of the parameters used for the hyperparameter search, with the only exception of the number of training steps, which we increased to 100,000 and we could see the loss had converged and remained stable after 60,000 (which is when the linear decay stops).

## D Experiments under Deterministic Target Policy

Our experiments in the 2D navigation environment so far used human demonstrations for all tasks. Human demonstrations are naturally stochastic which might add further complexity to learning a policy from these demonstrations in addition to the stochastic transitions of the environment. In this section, we conduct additional experiments within the same four tasks but with policies being trained on demonstrations collected from a deterministic A$^*$ planner.

### D.1 Data Collection and Dataset Details

**A$^*$ planner.** Given a state, the A$^*$ planner computes an optimal plan to the next unreached goal and executes the first action along this plan. We note that this planning process is executed under average transitions which are noise-free since the Gaussian noise added within the transition function of the environment has zero mean (see Appendix B.1 for more details). To ensure that the planner is able to react to noise, we re-compute the plan to the next goal at every step, as in Receding Horizon Control (Kwon & Han, 2005).

**A$^*$ datasets.** Table 6 shows statistics for each 2D navigation task and the collected A$^*$ planner datasets. We also visualize the 50 collected demonstrations of the human and A$^*$ planner for each of the tasks in Figure 8. From this visualization, we can see that the A$^*$ demonstrations tend to exhibit significantly lower variance in their trajectories, a trend that is particularly apparent in the more complex Maze and Multiroom tasks, leading to a more narrow state visitation distribution.

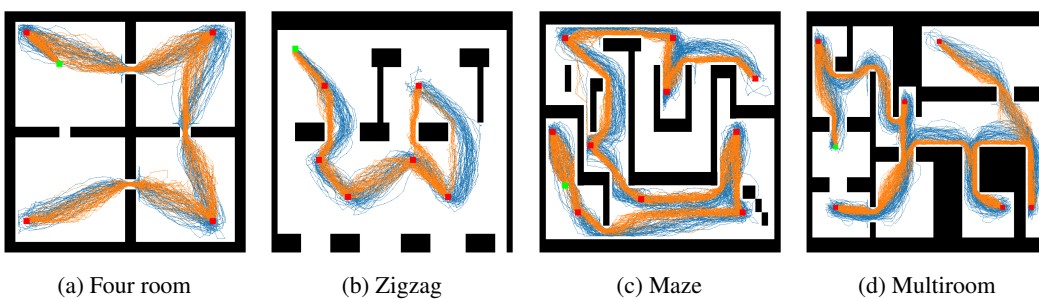

| (a) Four room | (b) Zigzag | (c) Maze | (d) Multiroom |

Figure 8: Traces of the 50 human (blue) and A* planner (orange) trajectories within all four 2D navigation tasks.

Table 6: Statistics of all four 2D navigation tasks and the A* planner datasets. The first four columns correspond to properties of the tasks, given by the number of goals, maximum number of time steps to complete the task, and the state dimensionality, while the last four columns correspond to the total number of trajectories/ time steps within the collected dataset (across all 50 trajectories) and statistics over the trajectory length.

| Task | Num goals | Max time steps | $|s|$ | Total steps | Trajectory length Min | Avg | Max |
|---|---|---|---|---|---|---|---|
| Four room | 4 | 200 | 14 | 6148 | 115 | 122.96 | 131 |
| Zigzag | 6 | 150 | 20 | 3820 | 71 | 76.4 | 85 |
| Maze | 10 | 300 | 32 | 9777 | 186 | 195.54 | 207 |
| Multiroom | 6 | 500 | 20 | 13 146 | 245 | 262.92 | 277 |

## D.2 HYPERPARAMETER SEARCH FOR A* DATA

Similar to the hyperparameter tuning on human datasets for 2D navigation tasks (Appendix B.3), we observe that BC is less stable and more sensitive to learning rate variations when trained on A* planner demonstrations. To improve stability and evaluation robustness, we performed a search over 10 learning rate configurations in the Multiroom task using a training dataset of 50 demonstrations. These configurations were chosen based on those that yielded the highest evaluation performance in our original tuning for human datasets.

The best result for BC in Multiroom was achieved with a linear decay from $1e^{-3}$ to $1e^{-6}$ over 100 000 steps without gradient clipping. However, BC remained less robust on A* demonstrations. In contrast, PIDM showed stable performance across learning rates, so we used the same constant rate of $1e^{-5}$ as in the experiments with the human dataset.

To ensure reliable results despite BC's instability, we trained and evaluated each algorithm with 50 random seeds, compared to 20 seeds for the human data evaluation.

## D.3 EVALUATION RESULTS FOR 2D NAVIGATION WITH DETERMINISTIC TARGET POLICY

**PIDM with human vs A* demonstrations.** To assess the impact of the narrower data distribution of the A* planner datasets on PIDM, we follow the methodology described in Section 5. For each task, PIDM is trained using 1, 2, 5, 10, 20, 30, 40 and 50 randomly sampled demonstrations for 50 random seeds. We evaluate four checkpoints throughout training per seed using 50 rollouts and report aggregate performance for the best checkpoint of each task and number of training demonstrations.

Figure 9 compares PIDM's evaluation performance in the four 2D navigation tasks when trained on human vs A* planner datasets. PIDM models trained on A* demonstrations are notably more sample efficient, achieving high performance with as few as just one training demonstration. Table 7 further

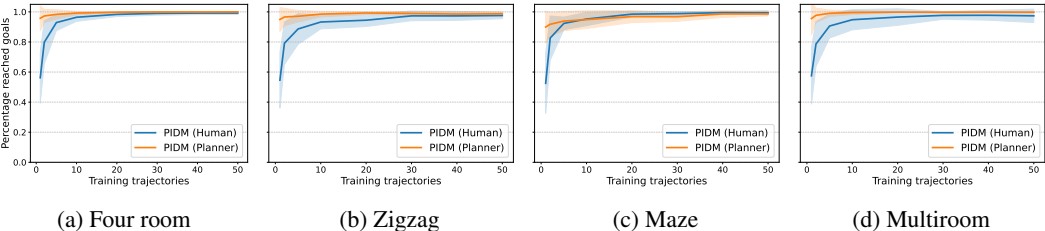

(a) Four room     (b) Zigzag     (c) Maze     (d) Multiroom

Figure 9: Performance per number of training demonstrations for PIDM in four tasks trained on human and A$^*$ planner demonstrations. Lines and shading correspond to the average and standard deviation across 20 and 50 seeds, respectively.

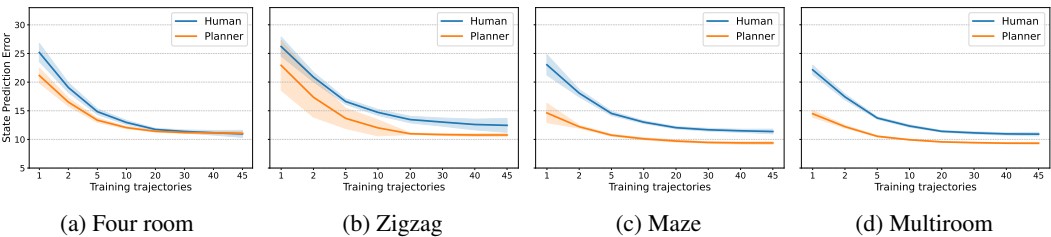

(a) Four room     (b) Zigzag     (c) Maze     (d) Multiroom

Figure 10: Error of the 2D navigation state predictor (as defined in Equation (15)) on held-out trajectories when trained on the human and A$^*$ planner demonstrations. Lines and shading correspond to the average and standard deviation across 50 seeds that determine the sampling of demonstrations used for training, respectively.

highlights this trend by showing the sample efficiency ratios between PIDM trained on the A$^*$ planner and human datasets.

**State predictor error.** Why is PIDM notably more efficient when trained on the narrower data distribution of A$^*$ planner demonstrations compared to human demonstrations? We hypothesize that the instance-based state predictor (see Equation (15)) provides more accurate future state predictions when trained on A$^*$ data. For states in held-out A$^*$ trajectories, the predictor is more likely to find similar states in the training set. Additionally, the A$^*$ collection policy is deterministic, resulting in lower action variability than the human policy. To investigate this hypothesis, we compute the state predictor error compared to ground-truth future states in held-out trajectories for each dataset and varying number of demonstrations. We consider multiple sizes $n \in \{1, 2, 5, 10, 20, 30, 40, 45\}$. For each $n$, we randomly sample 50 different subsets of the 50 available trajectories that are used to learn the state predictor (i.e. the instance-based state predictor defined in Equation (15) will lookup closest states and predicted future states within these trajectories), and use the remaining $(50 - n)$ demonstrations as held-out. From these held-out trajectories, we take all states and predict a future state with the state predictor and compare the predicted future state with the ground-truth.

Table 7: Maximum reached goal ratio and sample efficiency ratios of PIDM trained on A$^*$ planner demonstrations over PIDM trained on human demonstrations for 2D navigation tasks and average across tasks.

| Task | Four room | Zigzag | Maze | Multiroom | Average |
|---|---|---|---|---|---|
| max PIDM (Planner) $\uparrow$ | 1.00 | 0.99 | 0.99 | 1.00 | – |
| max PIDM (Human) $\uparrow$ | 0.99 | 0.98 | 0.99 | 0.98 | – |
| $\eta_{\text{PIDM(Planner)}}(80\%) \uparrow$ | 5.0 | 5.0 | 2.0 | 5.0 | 4.25 |
| $\eta_{\text{PIDM(Planner)}}(90\%) \uparrow$ | 5.0 | 10.0 | 2.5 | 5.0 | 5.625 |
| $\eta_{\text{PIDM(Planner)}}(95\%) \uparrow$ | 10.0 | 15.0 | 0.5 | 20.0 | 11.375 |

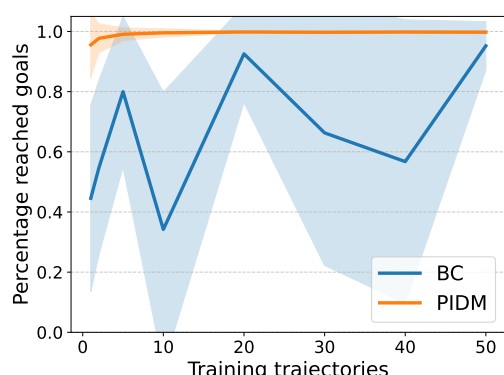

Figure 11: Sample efficiency of PIDM and BC in Multiroom, trained on A$^*$ planner data.

Figure 10 visualizes the state prediction error vs number of training demonstrations for each task and human and A$^*$ datasets. As expected, the state prediction error decreases as the number of demonstrations grows, but plateaus before 45 demonstrations. Crucially, the state state predictor error is notably lower for A$^*$ data, especially with few demonstrations, where the gap is largest. This reduced error correlates with the higher sample efficiency observed on the A$^*$ planner dataset (see Figure 9), providing empirical evidence for the effect of state predictor bias predicted by theoretical analysis, even in the small-data regime.

**PIDM vs BC trained on A$^*$ demonstrations.** In Section 5.3 we showed that PIDM is notably more efficient than BC when trained on human demonstrations. Figure 9 further demonstrates that PIDM achieves even greater sample efficiency when trained on the A$^*$ dataset, and this is correlated with the lower bias of the state predictor in this setting. Does BC similarly benefit similarly from the narrower distribution of A$^*$ data?

Figure 11 compares the sample efficiency for PIDM and BC in the Multiroom task when trained on A$^*$ data, aggregated across 50 random seeds. While PIDM clearly benefits, BC is negatively affected by the narrow data distribution, resulting in less stable training and overall lower sample efficiency compared to BC trained on the human data.

