# OpenReview forum: "When does Predictive Inverse Dynamics Outperform Behavior Cloning? Exploring the Role of Action and State Uncertainty"
_ICLR.cc/2026/Conference — Submitted to ICLR 2026_

### Official Review · Reviewer_7Wfv · 2025-10-18

**Soundness:** 2
**Presentation:** 3
**Contribution:** 2
**Rating:** 4
**Confidence:** 4

**Summary:**

This paper presents a thorough theoretical and empirical analysis of Predictive Inverse Dynamics Models (PIDMs) in the context of offline imitation learning, specifically contrasting them with the standard approach of Behavior Cloning (BC). The central research question is to explain ​​why and under what conditions PIDM achieves superior sample efficiency compared to BC​​, particularly in low-data regimes where expert demonstrations are scarce. The authors' primary contribution is a ​​novel theoretical framework​​ that quantifies the advantage of PIDM. They demonstrate that, under the assumption of a perfect state predictor, the expected prediction error for actions in PIDM is provably lower than or equal to that of BC. The theory further establishes a direct connection between this error reduction and sample efficiency gains, showing that PIDM can achieve a given performance level with fewer demonstrations.

**Strengths:**

1. A primary strength of this paper is its significant contribution to moving beyond empirical observation to a ​​principled theoretical understanding​​ of Predictive Inverse Dynamics Models (PIDMs). The authors provide a clear and compelling theoretical framework that explains why PIDM outperforms Behavior Cloning (BC) in sample efficiency.
2. Another strength of this paper lies in its ​​exceptionally thorough and multi-faceted experimental validation​​, which provides compelling, direct evidence for the theoretical claims. The empirical strategy is meticulously designed to test the theoretical predictions under increasingly general conditions, creating a powerful and convincing evidence chain.

**Weaknesses:**

While the paper makes a valuable contribution by providing a theoretical foundation for PIDMs, a significant weakness lies in the ​​limited novelty of its methodological contribution​​. The core architecture under study—decomposing policy learning into a future state predictor and an inverse dynamics model—is not a new invention but a re-investigation of an existing and increasingly popular paradigm. The paper itself positions the work as an explanation for "recent work", acknowledging that it is analyzing a pre-existing class of methods. The paper's primary contribution is thus ​​retrospective analytical justification​​ rather than ​​prospective algorithmic innovation​​.

The goal-conditioned or future-conditioned imitation learning paradigm, which PIDM embodies, has been extensively explored in various forms, such as in goal-conditioned behavior cloning, hindsight experience replay, and other methods that leverage future state information to guide policy learning. The paper's approach of using a state predictor (which is analogous to an action-free forward model) and an IDM policy is a valid but well-established instantiation of this idea. The authors do not propose a novel architecture, a more efficient training scheme, or a significant modification to the existing PIDM framework. Instead, they offer a rigorous theoretical explanation for why an existing approach works well, which, while intellectually valuable, does not advance the methodology itself.

**Questions:**

Is it possible to validate PIDM's performance on standard continuous control benchmarks, such as MuJoCo locomotion tasks?

---

> ### Author Response · Authors · 2025-11-24
> **Rebuttal Response**
>
> We thank the reviewer for their comments. We are encouraged that they find our work to be well presented, that our analysis clearly explains why PIDM outperforms BC in sample efficiency, and that they consider our empirical experiments to be meticulously designed. Below, we respond to their questions and concerns.
>
> ## Q: Comparison in standard control benchmarks
>
> We would like to emphasize that prior work has already provided substantial evidence for the claims that PIDM approaches can perform better than BC in complex control tasks, including continuous control benchmarks such as LIBERO, Robomimic, and ALOHA robotic manipulation tasks. We refer the reviewer to our common response on “Lack of standard/ more complicated benchmarks” for more details on existing empirical experiments of prior work. Furthermore, while these prior works have already shown that PIDM can perform better than BC in complex control tasks, it remained unclear when and why PIDM can obtain such performance gains over BC. Our work fills this gap by answering these questions through theoretical analysis and complementary empirical experiments.
>
> ## W: Lack of novel architecture or methodology
>
> We appreciate that the reviewer recognizes our primary contribution to be our analysis and understanding rather than a novel architecture or algorithm. However, the reviewer appears to still question the significance of our work given it “does not advance the methodology itself”.
>
> We believe that the analysis and novel insights provided by our work are valuable contributions for the imitation learning community. In fact, we would argue that the existence of prior work that has already shown notable benefits of PIDM over BC (see common response on “Lack of standard/ more complicated benchmarks”) further establishes the relevance of our insights as the derived efficiency gains clearly manifest in current state-of-the-art imitation learning algorithms.
>
> Beyond advancing the understanding of current solutions, we believe that our analysis is also relevant for imitation learning practitioners. Our theoretical analysis establishes conditions under which PIDM is guaranteed to be at least as sample efficient than BC, informing algorithm choice by establishing when benefits of PIDM over BC can be expected. Specifically, we show that PIDM should be preferred when the state predictor is (asymptotically) unbiased or when its bias is smaller than the conditional variance.
>
> New experiments with data from a deterministic A* planner (Appendix D), where we find the state-predictor error to be lower than in the human data case, showed even greater sample efficiency gains than with human data. These results suggest that the state predictor bias is the major driver for efficiency gains, as predicted by the new Theorem 3. Hence, as long as we can reasonably predict the future state (note our experiments rely on very simple instance-based state predictors), PIDM should be given a chance.

---

### Official Review · Reviewer_jyVk · 2025-10-27

**Soundness:** 2
**Presentation:** 3
**Contribution:** 2
**Rating:** 4
**Confidence:** 4

**Summary:**

This work aims to study the reason why predictive inverse dynamics models (PIDMs) outperform behavior cloning (BC), even in low-data regimes. The core contribution lies in the theoretical analysis, which shows that conditioning on future states reduces action uncertainty, leading to improved sample efficiency.

**Strengths:**

1. The paper discusses a novel perspective on the PIDM framework, offering insights into *why* PIDM can outperform BC, particularly under low-data regimes. Unlike prior work that primarily focuses on scaling up the training of state predictors, this work shows the fundamental role of *uncertainty reduction* in explaining PIDM’s improved sample efficiency.
2. By simplifying high-dimensional inputs to a 2D navigation setup, the analysis effectively isolates the key factors contributing to PIDM’s advantage. This design disentangles the benefits of learned representation quality from those of the decomposed learning paradigm, showing that PIDM’s gains arise from reduced action uncertainty when conditioning on future states.

**Weaknesses:**

Overall, the title suggests a broad discussion of *“when”* PIDM outperforms BC, but the analysis is conducted under a much narrower and idealized setting.

1. The theoretical finding — that conditioning on future states can reduce the uncertainty of predicting actions — is intuitive: if the policy knows the goal, predicting the corresponding action naturally becomes easier. This, however, transfers the difficulty from predicting actions to predicting future states. The paper does not analyze the relative complexity of these two estimation problems or provide conditions under which the proposed advantage provably holds.
2. The theoretical framework assumes near-perfect state prediction, an assumption that rarely holds in real-world, high-dim environments — especially under the low-data regime emphasized by the authors. In practice, imperfect state predictors cause the IDM to be conditioned on OOD future states and leading to compounding error. The paper would benefit from a discussion of how state-prediction errors affect the claimed theoretical guarantees.
3. The experiments are relatively simple, even in the 3D game environments, and do not consider settings with high action complexity. Prior work has shown that tasks involving precise control (e.g., dexterous manipulation in robotics) often expose the limits of models that rely on accurate future-state prediction. A more systematic empirical study — examining how state and action complexity influence PIDM’s advantage over BC — would be necessary to substantiate the paper’s general claims.

I encourage the authors to either reframe their contribution around the idealized theoretical setting or extend the analysis to address the above concerns. I will carefully reconsider my recommendation after the rebuttal.

**Questions:**

1. How does varying the number of future steps $k$ used during inference affect the results?

2. In the 2D environment, how large is the state-prediction error (or variance) in regions with high $\Delta(s)$?

---

> ### Author Response · Authors · 2025-11-24
> **Rebuttal Response**
>
> We thank the reviewer for their comments. We are encouraged that they find our work to be well presented and effectively isolates the key factors contributing to PIDM’s advantage. Below, we respond to their questions and concerns.
>
> ## Q1: Impact of varying $k$ during inference
>
> At inference time, we find that a lower k (as long as it is included in the set of k’s used for training) typically results in better rollout performance. We hypothesize that this is because, as $k$ increases, the variance of the predicted future state $s\_{t+k}$ also increases, leading to higher state prediction error and, thus, reduced performance (as predicted by our theoretical analysis).
>
> To validate this hypothesis, we approximate the variance $Var\_{s\_t}(s\_{t+k})$ of future states for varying $k$’s on the human datasets in all four 2D navigation tasks using the same clustering methodology used for our analysis in Section 5.4. As shown in the table below, the variance of future states substantially increases with $k$.
>
> | Task       | $k=1$  | $k=5$  | $k=10$  | $k=20$  | $k=50$   |
> |------------|--------|--------|---------|---------|----------|
> | Four room  | 142.17 | 505.91 | 1184.06 | 4144.00 | 8356.60  |
> | Zigzag     | 148.36 | 678.74 | 1137.68 | 4005.24 | 7635.88  |
> | Maze       | 187.55 | 592.80 | 1106.19 | 2763.16 | 7007.15  |
> | Multiroom  | 203.38 | 573.24 | 1080.78 | 2542.16 | 6547.84  |
>
> This variance is caused by the stochastic transitions of the environment with noise accumulating over multiple steps, and the stochastic action selection of human play and will negatively affect the accuracy of the state predictor. As demonstrated in our experiments comparing PIDM trained on human and A* planner data in Appendix D.4, this increased state-prediction error is also expected to lead to worse rollout performance of PIDM.
>
> ## Q2: State prediction error analysis
>
> In Appendix D of the revised submission, we added new analysis of the state prediction error for PIDM in the human dataset and a new A* planner dataset (Figure 10). Overall, we find that the error of the state predictor on held-out trajectories is lower in the planner dataset due to its narrower state distribution, and this lower state-prediction error results in improved online evaluation performance of PIDM when trained on the planner dataset.
>
> Regarding the correlation between the state-prediction error and $\Delta(s)$, we note that the A* policy is deterministic and only conditioned on the current state, so $\Delta(s) = 0$ in all states for the A* dataset. We hypothesize that this reduction in action variance, and the resulting narrower state distribution, is the main reason for the state prediction error in the A* dataset being lower compared to the human dataset where $\Delta(s) > 0$ in key states, as shown in Section 5.4.
>
> ## W1 & W2: Analysis for imperfect state predictors and when sample efficiency gains provably hold
>
> We have extended the theoretical analysis to consider any estimator, both biased and noisy estimators (Corollary 1 and Theorem 2). One key additional insight resulting from this extension is that the PIDM architecture introduces a bias and variance tradeoff: while the state predictor introduces bias due to a data distribution shift at test time, conditioning the IDM on the predicted state reduces variance. Furthermore, we derived a new Theorem 3 and Corollary 2 that provide the conditions under which PIDM is provably guaranteed to be more sample efficient than BC.
>
> Finally, in Appendix D.3 of our revised submission, we empirically validate these theoretical findings by analysing the state prediction error and its impact on the sample efficiency of PIDM. We find that state predictors trained on deterministic A* planner demonstrations exhibits lower state prediction error compared to models trained on human demonstrations. This lower state prediction error results in further improvement of the sample efficiency gains of PIDM when trained on the A* planner demonstrations compared to PIDM trained on human demonstrations, indicating that even in the low-data regime, a low state prediction bias and resulting sample efficiency of PIDM can be achieved.
>
> ## W3: Evaluate in benchmarks with higher state and action complexity
>
> We would like to emphasize that prior work has already provided substantial evidence for the claims that PIDM approaches can perform better than BC in complex control tasks, including complex robotic manipulation tasks, under identical numbers of samples. We refer the reviewer to our common response on “Lack of standard/ more complicated benchmarks” for more details on existing empirical experiments of prior work. Furthermore, while these prior works have already shown that PIDM can perform better than BC in complex control tasks, it remained unclear when and why PIDM can obtain such performance gains over BC. Our work fills this gap by answering these questions through theoretical analysis and complementary empirical experiments.

---

### Official Review · Reviewer_21mo · 2025-10-28

**Soundness:** 2
**Presentation:** 3
**Contribution:** 1
**Rating:** 2
**Confidence:** 4

**Summary:**

This paper presents a theoretical analysis explaining when and how **predictive inverse dynamics models (PIDM)** can outperform **behavior cloning (BC)** in offline imitation learning. They show that the conditioning on future states reduces the prediction variance of the policy, which benefits the sample efficiency of PIDM compared to BC.

**Strengths:**

**Motivation:** The paper tackles a relevant and underexplored question — why predictive inverse dynamics (PIDM) can outperform behavior cloning (BC) in offline imitation learning.

**Clarity:** The analysis is mathematically sound and well-structured, deriving intuitive results.

**Coherent:** The experimental results align well with the theoretical predictions.

**Weaknesses:**

- The theoretical analysis assumes point estimators, whereas recent imitation learning approaches (e.g., *Diffusion Policy*, *Flow Matching*) model full action distributions. Including a diffusion-policy baseline or an experiment using a distributional variant of PIDM would strengthen the paper’s practical relevance and clarify whether the predicted sample-efficiency advantage persists under more expressive policy classes.

- Equation (9) indicates that the efficiency gain depends on the stochasticity of the environment and the stochasticity of the expert policy. The current experiments introduce environmental noise but do not disentangle these two factors. Including controlled experiments that vary each source of randomness independently would provide valuable insight into how much of the observed efficiency gain is due to environmental stochasticity versus demonstration variability.

- The experimental environments are limited to navigation tasks, while imitation learning is widely applied to domains such as manipulation, locomotion, and autonomous driving. I would strongly encourage the authors to include a more diverse set of environments and tasks to demonstrate the generality of the assumptions.

- On page 8, the same symbol $k$ is used for both the prediction horizon ("$k$-step") and the number of clusters ("$k$-means"). While these meanings are unrelated, the reuse of notation can be confusing to readers. I would recommend adopting distinct notations.

**Questions:**

One open question is how practitioners can determine when to prefer PIDM over standard behavior cloning in practice. The current analysis shows that PIDM outperforms BC when future-state conditioning reduces action uncertainty, but it remains unclear how a user can recognize such conditions in a new domain. Could the authors provide more practical guidance or diagnostic criteria (e.g., measurable indicators of state–action ambiguity or future predictability) that would help a user decide when PIDM is expected to yield a tangible benefit?

---

> ### Author Response · Authors · 2025-11-24
> **Rebuttal Response**
>
> We thank the reviewer for their comments. We are encouraged that they consider the question studied in our work both relevant and underexplored, and found our analysis well-structured and sound. Below, we respond to their questions and concerns.
>
> ## Q: Practical guidance for practitioners to choose between PIDM and BC
>
> Our new theoretical analysis establishes conditions on the state predictor bias that guarantee PIDM’s sample efficiency is at least as high as that of BC. Specifically, PIDM should be preferred when the state predictor is (asymptotically) unbiased or when its bias is smaller than the conditional variance.
>
> New experiments with data from a deterministic planner (Appendix D), where we find the state-predictor error to be lower than in the human data case, showed even greater sample efficiency gains than with human data. These results suggest that the state predictor bias is the major driver for efficiency gains, as predicted by the new Theorem 3. Hence, as long as we can reasonably predict the future state (note our experiments rely on very simple instance-based state predictors), PIDM should be given a chance.
>
> Finally, while identifying scenarios with high conditional variance remains an open research question, we can offer some intuition: high conditional variance can arise in partially observable environments, where the current observation might be insufficient to discriminate between states where different actions are being taken, or perhaps in real-time human demonstrations, where delayed responses increase action uncertainty.
>
> ## W1: Point-estimator models
>
> Please see our common response on “more complex policy classes” where we explain how our new theoretical results provide a fundamental bias-variance tradeoff that goes beyond specific modelling choices, and how prior work has already established that PIDM under more complex model architectures (e.g. transformers or diffusion) can outperform BC under identical training conditions.
>
> ## W2: Disentangled stochasticity of environment and data collection policy
>
> We thank the reviewer for their suggestion. To disentangle the impact of stochasticity as part of the environment and the stochasticity of the expert data collection policy on the learning of BC and PIDM, we extended our experiments in the 2D navigation environment for new datasets collected by a deterministic A* planner. These experiments can be found in Appendix D of our revised submission.
>
> As expected, the A* planner leads to a narrower state distribution compared to the stochastic human data collection. We find that this change in data distribution affects PIDM and BC differently. While we find that PIDM benefits from the narrower data distribution and exhibits improved sample efficiency over PIDM trained on human demonstrations, we find that BC performs notably worse on this dataset.
>
> For PIDM, we find that the improved sample efficiency for the A* planner demonstrations is correlated with a lower state prediction error for this dataset compared to a state predictor trained on the human demonstrations. This lower state prediction error is the result of the narrower state distribution within this dataset.
>
> ## W3: Diverse set of environments and tasks
>
> While we agree that more empirical evidence can often be beneficial, we argue that this is not necessary given the contribution of our work. Recent work has already provided substantial evidence for the claims that PIDM approaches can perform better than BC in complex control tasks under identical numbers of samples. However, it remained unclear when and why PIDM can obtain such performance gains over BC. Our work fills this gap by answering these questions through theoretical analysis and complementary empirical experiments. For more details on existing empirical experiments of prior work, we refer the reviewer to our common response on “Lack of standard/ more complicated benchmarks”.
>
> ## W4: Notation for $k$
>
> Thanks for the careful reading. We agree this could cause unnecessary confusion. We have adopted capital $K$ for $K$-means.

---

### Official Review · Reviewer_5WJD · 2025-10-29

**Soundness:** 3
**Presentation:** 2
**Contribution:** 3
**Rating:** 6
**Confidence:** 3

**Summary:**

Predictive inverse dynamics models (PIDM) sometimes perform significantly better than Behavioral Cloning (BC), but the underlying reason is underexplored. This paper provides theoretical proof and insight for the sample efficiency of PIDM versus BC, including:
1. The prediction error of an optimal estimator for PIDM is always less than or equal to that of BC. The gap is characterized by the expected conditional variance of actions given future states, averaged over the current state distribution.
2. Under asymptotic efficiency assumptions, this uncertainty reduction can translate into sample efficiency gains.

The paper conducts visualization experiments on simple 2D tasks to support the points listed above and further validates these findings in a complex 3D environment, demonstrating significant sample efficiency gains over BC in both settings.

**Strengths:**

- The paper's main strength is its original theoretical analysis formally explaining why PIDM can be more sample-efficient than BC. It proves that PIDM's optimal prediction error is always less than or equal to BC's, providing a significant and principled reason to use PIDM in situations with high action uncertainty. This provides a strong theoretical backbone for a previously unexplained empirical observation.
- The experiment and validation are of high quality and clearly support the theory. The 2D navigation experiments are particularly clear, as the author visualizes the theoretical error gap and shows the PIDM policy learns to attend to future states only in these high-uncertainty regions. The validation in a complex 3D world confirms these findings are applicable to real-world tasks.

**Weaknesses:**

- The connection between Theorem 2 and efficiency gains in Section 4 is not entirely clear and easy to follow. Theorem 2 provides a valuable theoretical link based on an "asymptotic efficiency" assumption. However, the paper's main empirical results are in the "low-data regime" (e.g., 1-30 trajectories). The paper would be more convincing if it discussed whether these asymptotic assumptions are expected to hold in such low-data settings, or how this potential mismatch affects the practical gains observed.
- The scope of the comparison is narrow. The paper compares PIDM against a simple, point-estimator BC.

**Questions:**

- Around line 48, the Introduction states a focus on the "low-data regime" where "no additional data can be assumed", but then links this to the "current AI landscape, where large foundation models are trained on massive datasets". This connection is confusing. Could the authors please clarify how their setting, which explicitly assumes no additional data, is "increasingly relevant" to a landscape defined by massive pre-training? Is the intended link about the fine-tuning or alignment of these large models to new tasks, for which supervision is limited?
- In Theorem 2, the conclusion that the sample efficiency ratio $\eta \triangleq \frac{n}{m} \ge 0$ seems trivial, as $n$ and $m$ are sample counts and must be positive. The more practical and significant result would be $\eta \ge 1$ (i.e., PIDM is at least as sample-efficient as BC, $n \ge m$). Does the theory guarantee $\eta \ge 1$? Based on Equation (9), this seems to depend on the ratio $\frac{C_\mu}{C_\xi}$. Can the authors clarify the conditions under which $\eta \ge 1$ is guaranteed by the theory?

---

> ### Author Response · Authors · 2025-11-24
> **Rebuttal Response**
>
> We thank the reviewer for their comments. We are encouraged to see that they consider our analysis a “strong theoretical backbone for a previously unexplained empirical observation” and of high quality and clarity!
>
> Below, we address the reviewers questions and concerns.
>
> ## Q1: Connection to setting and relevance for foundation models
>
> The reviewer’s understanding is correct. Foundation models are pre-trained on large datasets, but they still require fine-tuning to solve many downstream tasks. This fine-tuning of foundation models often needs to be highly sample efficient as only few samples might be available for the downstream task.
>
> However, the low data regime is also important by itself in cases for which foundation models are not expected to generalize (e.g., industrial robotics with unusual sensing modalities or embodiments). To make this clearer, we have modified the sentence as follows:
>
> “In contrast to approaches that rely on extensive pretraining, we focus on the low-data regime, where only few demonstrations are available for the target task, and no additional data can be assumed. This setting is increasingly relevant in the current AI landscape, where large foundation models are trained on massive datasets, yet aligning them to new domains with limited supervision remains a significant challenge.”
>
> ## Q2: Theorem 2 clarifications
>
> We thank the reviewer for this comment. We have extended Theorem 2 to any estimator with the additional insight that the PIDM architecture introduces a bias and variance tradeoff: the state predictor introduces bias due to a data distribution shift at test time, while the IDM reduces variance by conditioning on the (predicted) future state.
>
> Then, we have added a new theorem and corollary with conditions that guarantee PIDM is at least as sample-efficient as BC. Theorem 3 is general, but with an intuitive insight: we just need the state predictor bias to be smaller than the variance reduction (assuming the bias functions change slowly). Corollary 2 customizes this result further and shows that an asymptotically unbiased state predictor guarantees non-negative sample efficiency gains.
>
> This new theorem and corollary rely on another new result that proves that the Fisher information ratio of BC over IDM is always equal or greater than one (see the new Lemma 2 in Appendix A.4).
>
> ## W1: Connection between Theorem 2 and empirical sample efficiency gains
>
> The asymptotic covariance often predicts finite-time performance well, as the Central Limit Theorem approximation is accurate for moderately large number of samples. For instance, the Berry–Esseen theorem states that, for i.i.d. samples, the convergence rate towards asymptotic normality is $1/ \sqrt{n}$, where $n$ is the number of samples,
>
> How this impacts our results is most clearly seen in the asymptotically unbiased case discussed in Corollary 2, where $\eta \ge  1$ holds because $\frac{F_\xi}{ F_\mu} \ge 1$ (see Lemma 2). Since the Fisher information is the inverse of the asymptotic covariance, this ratio implies that the IDM estimators' asymptotic covariance is no larger than that of BC. This is a fundamental fact in any data regime.
>
> Furthermore, we have extended both Corollary 1 and Theorem 2 to any (biased and noisy) estimator and both results show the same bias-variance tradeoff for EPE and sample efficiency: the variance reduction of the IDM increases the gap, while the state predictor introduces bias that reduces the gap. Combining the facts that Corollary 1 holds for any number of samples and that the fundamental mechanism is the same for both Corollary 1 and Theorem 2 suggests that Theorem 2 should also hold more generally than in the asymptotic regime.
>
> Indeed, Section 5 and Appendix D.3 provide empirical evidence of efficiency gains and of the role of the state predictor bias even in the small-data regime.
>
> ## W2: Point-estimator PIDM and BC
>
> Please see our common response on “more complex policy classes” where we explain how our new theoretical results provide a fundamental bias-variance tradeoff that goes beyond specific modelling choices, and how prior work has already established that PIDM under more complex model architectures (e.g. transformers or diffusion) can outperform BC under identical training conditions.

---

> > ### Comment · Reviewer_5WJD · 2025-11-27
> >
> > Thank you for the detailed clarifications and updates. All of my concerns are fully addressed.

---

> > > ### Author Response · Authors · 2025-11-28
> > >
> > > We thank the reviewer for their response. If all their concerns are fully addressed, we'd appreciate if they could update their rating, or let us know why they would still consider our work borderline.

---

### Official Review · Reviewer_MfP4 · 2025-10-31

**Soundness:** 3
**Presentation:** 3
**Contribution:** 2
**Rating:** 4
**Confidence:** 4

**Summary:**

This paper investigates why Predictive Inverse Dynamics Models (PIDM) outperform standard Behavior Cloning (BC) in offline imitation learning, particularly in low-data regimes. The authors theoretically show that decomposing policy learning into a state predictor and an inverse dynamics model reduces the expected prediction error relative to BC. The authors then provide empirical study to augment their theoretical contributions.

**Strengths:**

- Provides the formal theoretical result on why PIDM improves over BC.
- Proofs are simple and easy to follow.
- Empirical Study is provided on both 2D and 3D domains.

**Weaknesses:**

- The analysis does not quantify degradation when predictors are noisy or biased. At least Empirical tests under imperfect predictors would strengthen the overall idea. (Assumption of perfect or near-perfect state predictor)

- Missing comparison with recent BC variants that can also implicitly reduce uncertainty (e.g., implicit BC, diffusion BC) (Empirical limitation)

- Formal proof of theoretical results follow standard mathematical machinery with no novel analysis. (Novelty issue)

- Code is missing

**Questions:**

- In theorem 2, why does having $\eta \geq 0$ suffice? I thought $\eta \leq 1$ is what would make PIDM more sample efficient. Please correct me if there are any gaps in my understanding.

- In 2D experiment, you fix $k = 1$ whereas in 3D you train with multiple $k$ but evaluate at $k=1$. How sensitive are gains to $k$? Could you please sweep $k$ and report how the empirical $\Delta$ (and final performance) changes, balancing predictability of $s_{t+k}$ versus its disambiguation power?

- Your theory assumes access to a state predictor, while the experiments use non-learned predictors (instance-based in 2D; fixed-trajectory in 3D). Could you report results with a learned predictor (e.g., MLP/Transformer trained on $(s_t, s_{t+k})$​ to quantify how imperfect ‘future state distribution’ learning affects PIDM’s sample-efficiency advantage?

- Why is the code missing? Given that there are a number of experiments in the paper, how can I run and validate the claims? The anonymous code link or the zip file should have been provided in the supplementary file.

- Please fix minor formatting issues. For example, in line 107, after PIDM, ----- should not be there.

---

> ### Author Response · Authors · 2025-11-24
> **Rebuttal Response (1/2)**
>
> We thank the reviewer for their comments. We believe that by addressing them, the contribution of this work is stronger. Below, we respond to their raised questions and concerns.
>
> ## Q1: Theorem 2 understanding
>
> Your understanding is correct. We have now extended Theorem 2 in two ways. First, we extend the analysis to any estimator. Second, we added Theorem 3 and Corollary 2 that provide conditions for $\eta \ge  1$, guaranteeing PIDM’s sample efficiency to be greater than or equal to that of BC.
>
> ## Q2: Impact of varying $k$ for PIDM
>
> $k$ affects the PIDM algorithm differently at training and rollout times.
>
> At training time, as the literature on representation learning with IDM has shown (see Section 2 for discussion on related work on IDMs), having large enough $k$ can improve representations, e.g. by reducing the impact of exogeneous noise, under some assumptions. In practice, the more we expect representation learning to be crucial (e.g., having high dimensional vision inputs), the more we should consider sampling $k$ from a large range of values. However, very large $k$ can introduce variance, making training harder.
>
> At test time, $k$ should be chosen from those used during training to avoid distribution shift. In our experiments, we observe that using a smaller $k$ --- as long as it was included in the training set --- typically results in better rollout performance. We hypothesize that this is because, as $k$ increases, the variance of the predicted future state $s\_{t+k}$  also increases, leading to higher state prediction error and thus reduced performance (as predicted by our theoretical analysis).
>
> Below, we approximate the variance $Var\_{s\_t}(s\_{t+k})$ of future states and $\Delta$ (Equation 6) computed for varying $k$’s on the human datasets in all four 2D navigation tasks using the same clustering methodology used for our analysis in Section 5.4.
>
> $Var\_{s\_t}(s\_{t+k})$: As shown in the table below, the variance of future states substantially increases with $k$. This variance is caused by the stochastic transitions of the environment with noise accumulating over multiple steps, and the stochastic action selection of humans during data collection and will negatively affect the accuracy of the state predictor. This bias is also expected to lead to worse online performance of the PIDM model.
>
> | Task       | $k=1$  | $k=5$  | $k=10$  | $k=20$  | $k=50$   |
> |------------|--------|--------|---------|---------|----------|
> | Four room  | 142.17 | 505.91 | 1184.06 | 4144.00 | 8356.60  |
> | Zigzag     | 148.36 | 678.74 | 1137.68 | 4005.24 | 7635.88  |
> | Maze       | 187.55 | 592.80 | 1106.19 | 2763.16 | 7007.15  |
> | Multiroom  | 203.38 | 573.24 | 1080.78 | 2542.16 | 6547.84  |
>
> $\Delta$: As shown in the table below, $\Delta$ is comparably insensitive to $k$, which makes sense as our theory predicts that this variance term is largely a function of the data collection policy, in this case the human play.
>
> | Task       | $k=1$  | $k=5$  | $k=10$  | $k=20$  | $k=50$   |
> |------------|--------|--------|---------|---------|----------|
> | Four room  | 0.115  | 0.124  | 0.110   | 0.109   | 0.105    |
> | Zigzag     | 0.209  | 0.208  | 0.202   | 0.215   | 0.209    |
> | Maze       | 0.177  | 0.183  | 0.175   | 0.174   | 0.167    |
> | Multiroom  | 0.130  | 0.134  | 0.122   | 0.116   | 0.117    |
>
> ## Q3: Learned state predictors
>
> Thanks for raising this point. Our theory just assumes access to a state predictor, independent of how it has been obtained. We have extended our analysis  to take into account that any approximate state-predictor can introduce bias due to a data distribution shift (i.e., the IDM was trained conditioning on the actual future states, while it only has access to predicted states at test time). This provides a new insight: the key driver of efficiency gains is the bias introduced by an approximate state predictor, whether instance-based or model-based or a particular family. We have clarified this in the updated draft.
>
> In Appendix D of the revised submission, we have added new experiments that study the effect of the state prediction error on sample efficiency. The results show how a smaller prediction error results in higher sample efficiency, as predicted by our new theoretical results.
>
> We emphasize that previous studies (Xie et al., 2025; Tian et al., 2025) already showed that learning a state predictor with diffusion models or transformers achieved superior performance than BC. Our theoretical analysis thus explains those previous results, reinforcing the relevance of our insights.
>
> We relied on an instance-based predictor as a simple model that facilitates a fair comparison of BC and PIDM with similar model capacity and training compute budget being given to both approaches.

---

> > ### Author Response · Authors · 2025-11-24
> > **Rebuttal Response (2/2)**
> >
> > ## Q4 & W4: Code access
> >
> > We are committed to releasing the code for the algorithms and toy environment if the paper is accepted. However, we are unable to release the code for interacting with the video game, as it contains proprietary dependencies.
> >
> > ## Q5: Format Fix
> >
> > Thanks for the detailed reading, we fixed the formatting.
> >
> > ## W1: Analysis and empirical experiments for noise or biased state predictors
> >
> > We have extended the theoretical analysis to any estimator (including biased and noisy estimators) for both EPE and sample efficiency. One key additional insight resulting from this extension is that the PIDM architecture introduces a bias and variance tradeoff: while the state predictor introduces bias due to a data distribution shift at test time, conditioning the IDM on the predicted state reduces variance.
> >
> > All the experiments were performed under realistic conditions, using a biased instance-based state predictor and noisy neural networks trained with limited data for the policy. The results show that the variance reduction outweighs the additional bias, leading to sample efficiency gains in both the 2D and 3D environments.
> >
> > ## W2: Comparison to BC variants (e.g. diffusion BC)
> >
> > Please see our common response on “more complex policy classes” where we explain how our new theoretical results provide a fundamental bias-variance tradeoff that goes beyond specific modelling choices, and how prior work has already established that PIDM under more complex model architectures (e.g. transformers or diffusion) can outperform BC under identical training conditions.
> >
> > ## W3: Novelty concerns for theoretical analysis
> >
> > While we agree that our theoretical analysis does not introduce novel techniques, our insights are novel and valuable and we believe their simplicity makes them more easily accessible to other researchers that might want to build on our work.
> >
> > For example, note that beyond robotics and gaming, our analysis applies more generally to BC losses, e.g. as applied in LLMs where the state is represented by the hidden state before the head and the action space is the token vocabulary.
> >
> > Additionally, to extend our theoretical analysis, we required Lemma 2, which combines several techniques to compare the Fisher information for IDM and BC estimators.

---

### Author Response · Authors · 2025-11-24
**Common Rebuttal Response (1/2)**

We thank all the reviewers for their feedback, and we are encouraged that they consider our work sound, well presented, and intuitive. Their comments helped us to strengthen our work with new revisions being highlighted in blue color within the submission PDF. In summary, these are the main changes:
- **Extended theory for imperfect future state predictors (Corollary 1 and Theorem 2)**: we extended the analysis of the prediction error and sample-efficiency gaps for PIDM with imperfect state predictors, as requested by the reviewers. This analysis provides a deeper understanding on PIDM: the architecture introduces a bias-variance tradeoff, where conditioning the policy on future states can reduce the variance of the policy estimator (shown now under realistic conditions) but an approximate state predictor may introduce bias.
- **New conditions that guarantee PIDM to be at least as sample efficiency as BC (Theorem 3 and Corollary 2)**: based on the extended sample efficiency analysis of Theorem 2, we derive conditions on the state predictor bias under which PIDM’s sample efficiency is guaranteed to be greater than or equal to that of BC.
- **New experiments to disentangle stochasticity of environment and data collection policy (Appendix D)**: we conducted new experiments in the 2D navigation environment in which we trained PIDM and BC on a dataset collected by a deterministic A* planner. We find that the resulting narrower state distribution significantly benefits PIDM while BC appears to suffer compared to the human datasets.
- **New analysis of the state prediction error (Appendix D.3)**: we quantify the error of the state predictor on held-out data for both the human and A* datasets and show that lower state prediction error is correlated with higher performance, as predicted by our theory.
- **Increased number of seeds in Figure 4 from 10 to 20 seeds** for more statistical confidence, and adjusted results in Figure 4 and Table 1 that show increased sample efficiency gains compared to the previously reported results for 10 seeds.

Below, we address several questions raised by multiple of the reviewers.

## Theory is limited to perfect state predictors

Stated by reviewer MfP4, jyVk

We have extended the analysis to the case of arbitrary estimators, which has allowed us to study the impact of approximate state predictors. The result reveals a key advantage of the PIDM architecture: a bias-variance tradeoff. Conditioning on a future state reduces the total variance by removing the conditional variance mentioned above; however, predicting a future state induces bias.

In particular, when we only have access to an approximate state predictor, denoted $\hat{p}$, the estimator of the IDM policy will generate actions conditioned on samples of the form $(s\_t, s'\_{t+k})$, with $s'\_{t+k} \sim \hat{p}(\cdot \mid s\_t)$ as opposed to from the true data distribution. This distribution shift introduces bias. In other words, the estimator of the IDM policy has two sources of bias: the bias that is intrinsic to the estimator and the additional bias due to the distribution shift. This additional bias reduces the predicted error and sample efficiency gaps, as shown in the updated version of Corollary 1 and Theorem 2.

## Lack of standard/ more complicated benchmarks

Stated by 21mo, jyVk, 7Wfv

We emphasize that recent empirical work already provides empirical support for the claims that PIDM can outperform BC in complex control tasks under identical number of samples. Below we refer to three recent works:
- **UniPi**: Du et al., 2023, “Learning Universal Policies via Text-Guided Video Generation”
  - **Approach**: PIDM with a diffusion model as state predictor.
  - **Empirical result**: PIDM (UniPi) is compared to BC with varying architectures (transformer vs diffusion policy) and conditioning sets in complex multi-task control tasks. This work shows that PIDM consistently outperforms all BC variants.
- **Seer**: Tian et al., 2025, “Predictive Inverse Dynamics Models are Scalable Learners for Robotic Manipulation”
  - **Approach**: PIDM as transformer model with joint end-to-end optimization of state predictor and IDM policy.
  - **Empirical result**: PIDM (named “Seer”) is compared to a transformer-based BC baseline in the LIBERO multi-task benchmark. Seer (PIDM) is shown to significantly outperform the BC baseline in 3 out of 4 tasks, even when they are trained with identical training data.
- **Latent Diffusion Planning (LDP)**: Xie et all, 2025, “Latent Diffusion Planning for Imitation Learning”
  - **Approach**: PIDM with a VAE as state predictor and a diffusion model for the IDM policy.
  - **Empirical result**: PIDM (named “LDP”) is compared to diffusion policy (DP), a diffusion model trained with BC, in Robomimic and ALOHA robotic manipulation tasks. Given the same data budget, PIDM (LDP) outperforms BC (DP) in 3 out of 4 tasks (Lift, Can, ALOHA Cube) and reaches comparable performance in the Square task.

---

> ### Author Response · Authors · 2025-11-24
> **Common Rebuttal Response (2/2)**
>
> Collectively, these results demonstrate that PIDM is broadly applicable and can exhibit performance gains over BC in a variety of complex control tasks. However, these previous works didn’t explain why PIDM can obtain such performance gains over BC. Answering this question through theoretical analysis and complementary empirical experiments is the main motivation for our work.
>
> ## No consideration for more complex policy classes (e.g. diffusion policy)
>
> Stated by MfP4, 5WJD (mentions only considering point-estimator BC), 21mo
>
> While our theoretical analysis focuses on point estimators, the underlying insight extends beyond specific modeling choices: behavior cloning (BC) can be viewed as the marginal of inverse dynamics modeling (IDM), which means BC inherently has greater uncertainty. This additional uncertainty should translate into efficiency gains for IDM, with the main limitation being the bias introduced when conditioning on an approximate state predictor.
>
> For a fair comparison, if a richer model is used to better capture the BC policy distribution $\pi_\mu(a_t | s_t)$, the same model should also be used for the IDM policy $\pi_\xi(a_t | s_t, s_{t+k})$. In this case, the uncertainty reduction from modeling multiple modes of the action distribution benefits both BC and IDM; however, IDM retains a fundamental advantage by further reducing action uncertainty through access to the (predicted) future state.
>
> Previous studies showed that PIDM outperforms BC with richer policy classes, like diffusion (Xie et al., 2025) or transformer (Tian et al., 2025) models, even when they use the same dataset. Those results provide evidence that the bias-variance we have unveiled is a fundamental feature of the PIDM architecture.
>
> Specifically for diffusion policies, we highlight the results of the work on latent diffusion planning (LDP) by Xie et al. (2025), which models both the state predictor and IDM policy of PIDM as diffusion models and compares them with diffusion policy (DP), a state-of-the-art BC-like approach to model action distributions. Even with identical number of demonstrations on all tasks, LDP achieves higher performance than DP. In ALOHA and Robomimic tasks, DP achieves an average success rate of 0.51, whereas LDP reaches 0.65. On Franka manipulation task, DP reaches 69.6% whereas LDP achieves 73.3%. These results demonstrate that even when both approaches rely on powerful diffusion models, the PIDM decomposition (future-state predictor + inverse dynamics model) yields substantially higher sample efficiency and downstream task performance compared to a single-stage distributional BC policy.
>
> So far, it remained unclear as to why PIDM is more effective than BC. Rather than repeating these experiments ourselves, we have focused on exploring the simpler case of point-based estimators for which we could develop theoretical insights that explain why PIDM can be more effective and performed empirical experiments that illustrate our theory. The fact that our analysis gives a natural explanation for results with richer policy classes reinforces its practical relevance. Our contribution is therefore not a new algorithm but a principled understanding of why the PIDM decomposition works and how it consistently improves learning efficiency in both simple and highly expressive policy settings.
>
> ## Theorem 2 appears rather limited
>
> Stated by MfP4, 5WJD
>
> We have extended Theorem 2 in two ways. First, we extend the analysis to any estimator. Second, we added new Theorem 3 and Corollary 2 that provide conditions under which PIDM is guaranteed to be more or equally sample efficient than BC.

---

> > ### Author Response · Authors · 2025-11-27
> > **Rebuttal Reminder**
> >
> > We would kindly ask the reviewers to take a look at our rebuttal response and the revised submission.
> > If your concerns have been addressed, we would appreciate if you could update your scores to reflect that. If you believe that your concerns remain, please let us know why you believe our rebuttal is insufficient at addressing your concerns so we can engage in discussion.
> >
> > Many thanks!

---

### Meta-Review · Area_Chair_MxKq · 2026-01-05

**Summary:**

This paper provides a theoretical study to answer the question why Predictive Inverse Dynamics Models (PIDM) outperform Behavior Cloning (BC) for offline imitation learning, especially in the low expert data regime. The authors characterize the bias-variance tradeoff induced by the future state predictor, and further identify the conditions on state predictor bias for PIDM being more sample efficient than BC.

Strengths:
- Provides the first formal theoretical analysis to explain why PIDM improves over BC

- Experiments on both 2D and 3D domains are conducted to support the theoretical findings

Weaknesses:
- The theoretical analysis only considers point estimators

- The experimental verifications are relatively simple and do not consider more complex benchmarks

**Reviewer Concerns:**

The authors' rebuttal has addressed several concerns raised by the reviewers, including an extension to the case with imperfect state predictors, additional experimental results to investigate the impact of $K$, state predictor error, and system stochasticity.

Despite this, this paper would benefit from stronger theoretical analysis on more complex models and more comprehensive empirical evaluations. Particularly, while the authors argue that the empirical advantage of PIDM over BC has been shown on more complex policy classes and benchmarks in the literature, it remains important to verify whether the theoretical insights presented here still hold in these setups, in order to sufficiently claim that this paper offers a principled explanation for why PIDM outperforms BC in practice.

**Reviewer Scores:**

Reviewer MfP4 may maintain the score, due to the concerns on limited experiments and technical novelty.

Reviewer 5WJD may maintain the score, as the initial score was positive and the reviewer didn't indicate strong support to this paper.

Reviewer 21mo may maintain the score, because the two critical limitations (only consider point estimator and limited empirical evaluations) still persist after the rebuttal.

Reviewer jyVk shared similar concerns with Reviewer 21mo and may maintain the score

---

### Decision · Program_Chairs · 2026-01-26

Reject